# Collective photothermal bending of flexible organic crystals modified with MXene-polymer multilayers as optical waveguide arrays

Xuesong Yang[1], Linfeng Lan[1], Liang Li [2,3], Jinyang Yu[4], Xiaokong Liu [1], Ying Tao[4] ✉, Quan-Hong Yang [4], Panče Naumov [2,5,6,7] ✉ & Hongyu Zhang [1] ✉

The performance of any engineering material is naturally limited by its structure, and while each material suffers from one or multiple shortcomings when considered for a particular application, these can be potentially circumvented by hybridization with other materials. By combining organic crystals with MXenes as thermal absorbers and charged polymers as adhesive counter-ionic components, we propose a simple access to flexible hybrid organic crystal materials that have the ability to mechanically respond to infrared light. The ensuing hybrid organic crystals are durable, respond fast, and can be cycled between straight and deformed state repeatedly without fatigue. The point of flexure and the curvature of the crystals can be precisely controlled by modulating the position, duration, and power of thermal excitation, and this control can be extended from individual hybrid crystals to motion of ordered two-dimensional arrays of such crystals. We also demonstrate that excitation can be achieved over very long distances (>3 m). The ability to control the shape with infrared light adds to the versatility in the anticipated applications of organic crystals, most immediately in their application as thermally controllable flexible optical waveguides for signal transmission in flexible organic electronics.

The speed and amount of transfer of information by optical means is one of the cornerstones of the current societal development and will very likely guide the progress of humanity in the near future. Among the variety of materials that are being considered as optical signal transducers, dynamic organic crystals[1–7] that have evolved into 'crystal adaptronics', an emerging field in materials science research[8], have only recently garnered attention as an alternative to the traditional silica-based fiber-optics with low density, reduced scattering, flexibility[9–12], and polarizing ability[13,14] as some of their most favorable assets. The realization that organic crystals can be used as transmitters

[1]State Key Laboratory of Supramolecular Structure and Materials, College of Chemistry, Jilin University, 130012 Changchun, P. R. China. [2]Smart Materials Lab, New York University Abu Dhabi, PO Box 129188, Abu Dhabi, UAE. [3]Department of Sciences and Engineering, Sorbonne University Abu Dhabi, PO Box 38044, Abu Dhabi, UAE. [4]Nanoyang Group, Tianjin Key Laboratory of Advanced Carbon and Electrochemical Energy Storage, School of Chemical Engineering and Technology, Tianjin University, 300072 Tianjin, China. [5]Center for Smart Engineering Materials, New York University Abu Dhabi, PO Box 129188, Abu Dhabi, UAE. [6]Research Center for Environment and Materials, Macedonian Academy of Sciences and Arts, Bul. Krste Misirkov 2, MK–1000 Skopje, Macedonia. [7]Molecular Design Institute, Department of Chemistry, New York University, 100 Washington Square East, New York, NY 10003, USA. ✉ e-mail: yingtao@tju.edu.cn; pance.naumov@nyu.edu; hongyuzhang@jlu.edu.cn

of light has quickly led to attempts to fabricate crystal-based replicas of various commonly used optical devices and circuits[15,16], an effort that capitalizes on the mechanical adaptability of some organic crystals[17]. Yet another direction of the currently very active research pursuit in this field is the attempt to diversify the range of stimuli that can be used to achieve spatial control over the optical output, which can be delivered both in the visible (see below) and near-infrared (telecom) range[18,19]. The available approaches to reshape crystalline optical waveguides include the application of mechanical force[20], light[21,22], humidity[23], and magnetic field[24]. Specifically, UV light can be used to drive a photochemical reaction in the crystal, generating internal strain that translates into a bending moment[25,26]. Although these photochemically driven deformations can be quite significant in the extent of bending or curling, the processes of amplification of the strain caused by the photochemical reaction at a molecular scale to macroscopic bending are known to be kinetically inefficient, and such processes generally are not thought to qualify for actuating applications[27]. Moreover, the high energy of the UV light may generate chemically reactive species within the structure that could undergo side reactions, and over a prolonged period, they inevitably result in operational fatigue.

Much faster deformations have been recently demonstrated by using photothermal effects caused by local heating with light; however, they result in minuscule deformations, typically on the order of less than $1°$[28]. Considering that organic crystals are generally poor thermal conductors, in an attempt to achieve thermal control over the optical output, we resorted to exploring hybrid materials where they are coupled with an efficient thermal absorbent. Herein, we report a simple and efficient method for preparing hybrid organic crystals capable of responding to infrared light. The crystals were coated with a 2D compound from the MXene family, $Ti_3C_2T_x$ (T stands for surface terminal groups, including OH, O or F), which is known for its high efficiency in photothermal conversion[29] and has been widely applied for biological, medical, and other purposes[30–35]. Similar to the MXene, the hybrid organic crystals were found to be thermally absorbing and to undergo thermomechanical motion when exposed to infrared radiation. The photothermal actuation of the hybrid crystals reported here comes with multiple advantages over other methods that have been used for crystal deformation. First, since the absorber is the MXene, this approach does not require the absorption of radiation by the crystal itself, which circumvents the necessity for the crystal to absorb light (i.e., be photoreactive). Second, both high speeds and strong deformations can be achieved, and the crystal can be actuated by localized thermal excitation at a predetermined position. Finally, we demonstrate that hybrid crystals can be precisely actuated by excitation over very long distances (>3 m), which brings an added value to the potential of dynamic responsiveness of organic crystals for real-world applications, where they could be used as receivers for remote sensing, triggering, or actuation. Within a broader context, our study demonstrates that the thermal excitation can be applied to control optoelectronic elements made of these hybrid crystals, such as, for example, optical waveguides, and this concept can be expanded to control ordered arrays of optically waveguiding dynamic crystals.

## Results and discussion
### Preparation of the hybrid organic crystal
The MXene used here has a typical two-dimensional lamellar structure[36], which is often used to prepare transparent MXene multilayers[37]. The diluted aqueous suspension of the MXene appears black in color, corresponding to the strong absorption in its UV–vis spectrum between 300 and 900 nm, with a characteristic absorption peak around 760 nm (Supplementary Fig. 1). MXene multilayers were fabricated by using the layer-by-layer (LbL) assembly technique, by alternative deposition of positively charged polydiallyldimethylammonium (PDDA) and negatively charged MXene nanosheets onto a solid substrate[38]. This

resulted in multilayered structures (PDDA/MXene)$_n$, which can have an arbitrary number of deposited bilayers ($n$) and can be prepared on practically any type of water-insoluble substrate, including but not limited to glass, silicon, ceramics, metals, and plastics. In a typical case, a crystal of the organic compound 2,2′-((1$E$,1′$E$)−1,4-phenylenebis(ethene-2,1-diyl))dibenzonitrile (for convenience, hereafter referred to as **1**; Fig. 1a) coated with 5 bilayers, **1**@(PDDA/MXene)$_5$, had a thickness of about 200 nm and low roughness (average roughness, 20.1 nm) (Fig. 1b, c; Supplementary Fig. 2)[39]. The mechanical properties of crystal **1** were tested by a three-point bending test (Supplementary Fig. 3). The (PDDA/MXene)$_n$-coated crystalline hybrids are very transmissive to visible light, although expectedly, by increasing the number of deposited bilayers, the transmittance gradually decreases in the visible spectral region (≈400–780 nm, Fig. 1d, e)[40] due to increased thickness. For instance, **1**@(PDDA/MXene)$_5$ has very high transmittance of 71% at 550 nm (Fig. 1e). As shown in Fig. 1f, when exposed to infrared light, the temperature of the surface of **1**@(PDDA/MXene)$_5$ ($\Delta T$) rises with the increasing power of the infrared light up to $\Delta T =$ 61.2 °C under 744 mW infrared radiation, reaching a constant value within 10 s (Fig. 1g). **1**@(PDDA/MXene)$_5$@PDDA/poly(styrene sulfonate) (PSS) was selected to investigate the relationship between the thermal effect caused by the MXene and the surface temperature when the hybrid crystal is exposed to infrared light. As shown in Supplementary Figs. 4 and 5 and Supplementary Movie 1, heat is transferred from the irradiated area to the surroundings due to the favorable thermal conductivity of the MXene. This raises the temperature of the surrounding crystal surface until it becomes constant after a few seconds. Furthermore, **1**@(PDDA/MXene)$_1$@PDDA/PSS and **1**@(PDDA/MXene)$_5$@PDDA/PSS were selected to demonstrate the effect of MXene thickness on the photothermal effect. As shown in Supplementary Fig. 6, the temperature change of **1**@(PDDA/MXene)$_5$@PDDA/PSS was about five-fold that of **1**@(PDDA/MXene)$_1$@PDDA/PSS at the same power of the infrared light. These results indicate that **1**@(PDDA/MXene)$_5$ has excellent infrared light absorption properties.

Encouraged by these initial results with **1**, the preparation method was slightly modified and applied to centimeter-long slender elastic crystals of three other organic compounds: 9,10-dibromoanthracene, ($Z$)−2-([1,1′-biphenyl]−4-yl)−3-(anthracen-9-yl)acrylonitrile, and ($Z$)−3-(furan-2-yl)−2-(4-((($E$)−2-hydroxy-5-methylbenzylidene)amino)phenyl)acrylonitrile (for convenience, hereafter referred to as **2**, **3**, and **4**, respectively, Fig. 2a), which were obtained by using literature methods[23,41,42]. All these crystals are elastic and can be bent repeatedly into a U-shape without breaking. As shown schematically in Fig. 2b and Supplementary Fig. 7, nascent (as-crystallized) crystals of **2**−**4** were first uniformly coated with a mixture of PDDA and a negatively charged PSS layer of ca. 650 nm thickness. The surfaces of the resulting hybrid crystals, **2**−**4**@PDDA/PSS (hereafter, **2**−**4**@P), were then coated with a mixture of (positively charged) PDDA and (negatively charged) MXene layer of ca. 200 nm thickness and described as **2**−**4**@PDDA/MXene@PDDA/PSS (for convenience, hereafter, **2**−**4**@P²). Lastly, by using a needle tip, a 2 μm-thick layer of polyvinyl alcohol (PVA) with PSS, PVA/PSS, was deposited uniformly and rapidly along only one of the bendable faces and left to dry, a step that afforded a hybrid described as **2**−**4**@PVA/PSS@PDDA/MXene@PDDA/PSS (hereafter, **2**−**4**@P³). Since only one of the two wide faces of the crystal is coated with the polymer, when the polymer shrinks, it gives rise to a differential strain that translates into a bending moment. This is observed as the macroscopic bending of the hybrid crystal. In the hybrid crystal, the MXene functions as a photothermal converter, while the PVA/PSS layer functions as one of the two components of a bilayer strip that generates bending moment by expansion or

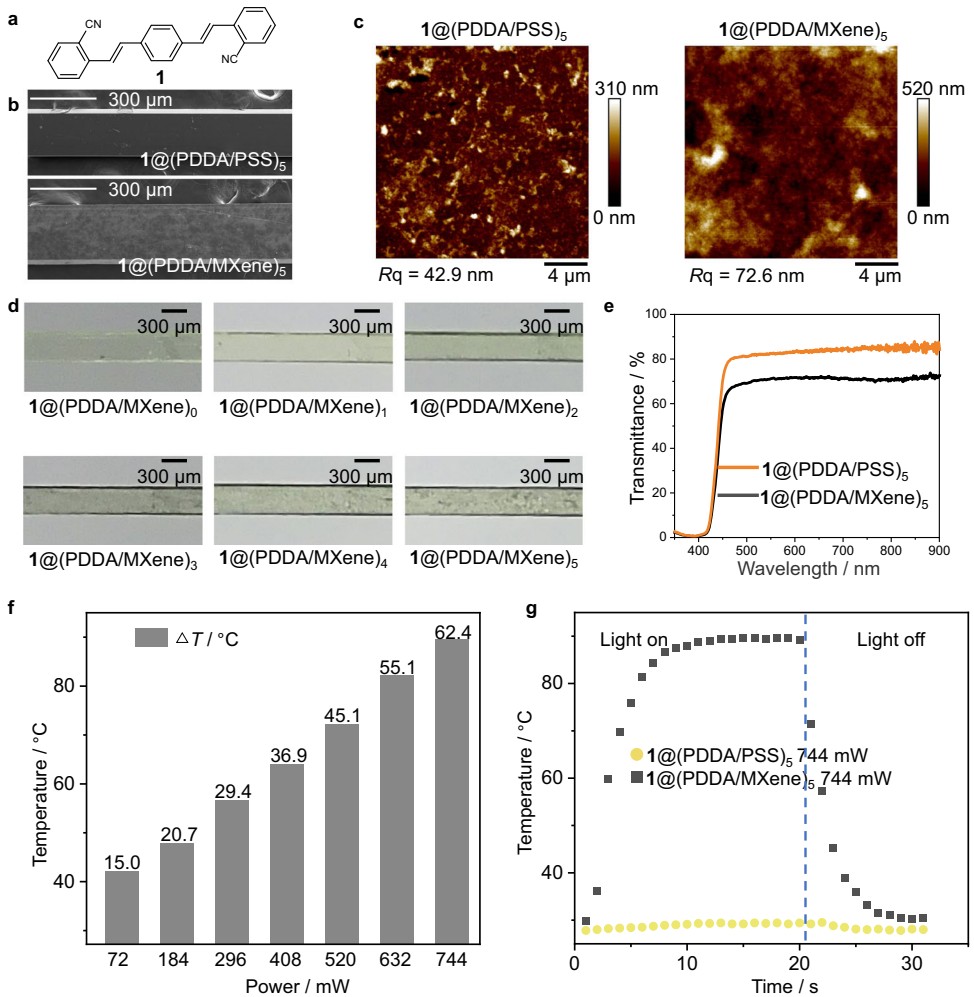

**Fig. 1 | Preparation and thermal properties of the MXene-polymer crystal hybrids. a** Chemical structure of **1**. **b** Scanning electron microscopy (SEM) images of the surfaces of **1**@(PDDA/PSS)$_5$ and the **1**@(PDDA/MXene)$_5$. The scale bar is 300 μm. **c** Atomic force microscopy (AFM) images of the surfaces of **1**@(PDDA/PSS)$_5$ and **1**@(PDDA/MXene)$_5$. $R$q is the root mean square value of the profile deviation from the mean over the sampling length. **d** Photographs of different MXene layers on the **1**@(PDDA/PSS)$_n$ surface. ($n$ represents the number of layers of MXene).

**e** UV–vis transmission spectra of **1**@(PDDA/PSS)$_5$ and **1**@(PDDA/MXene)$_5$. **f** Increase of the temperature of the coated crystal (PDDA/MXene)$_5$ with increasing power of the infrared radiation ($\Delta T$, 25 °C). **g** A time-dependent temperature increase of the surface of **1**@(PDDA/PSS)$_5$ and **1**@(PDDA/MXene)$_5$ upon illumination with infrared light (744 mW) at 25 °C. The dashed blue line separates the light regime (left) and the dark regime, i.e., absence of light (right). The source data is provided as a Source Data file.

contraction. PVA is a common hygroscopic polymer that has a low critical solubility temperature and is well known to undergo reversible swelling via hydrogen bond formation (Supplementary Fig. 8)[43]. This brings about a response of the hybrid element to water vapor, and the curvature could change with variation in humidity (Supplementary Fig. 9). Due to the presence of the MXene all these hybrid crystals can be bent by infrared radiation. As shown with the example of **4**@P³ in Supplementary Fig. 10 and Supplementary Movie 2, while the crystals of **4**@PVA/PSS@PDDA/PSS practically do not show a response to infrared radiation at 184 mW, **4**@P³ clearly bends under the same conditions. To further confirm that the bending of the hybrid crystals was a result of the thermally induced mechanical process and not a physical phase transition or a chemical reaction, NMR analysis of **2**–**4** was performed before and after heating at 100 °C for 1 h, and the compounds were also analyzed by differential scanning calorimetry (DSC) (Supplementary Figs. 11–14). The results ruled out phase transitions below 100 °C (note that one of the compounds undergoes a phase transition at 197 °C) or permanent chemical changes. In order to examine the effect of the coating with polymer and application of MXene on the mechanical properties,

the stress–strain profiles of the crystals **2**, **3**, **2**@P³, and **3**@P³ were compared (Supplementary Fig. 15). The results confirmed that the mechanical properties of the nascent crystals were essentially retained in the hybrid crystals, with a very small change in the Young's modulus. It is natural to expect that the crystal structure of the crystal determines its deformation. In line with this, as shown in Supplementary Fig. 16, the bending degrees of the hybrid crystals **2**–**4**@P³ are different under identical conditions of excitation. An additional, and perhaps less obvious factor that could affect the bending, is the crystal quality. To that end, a high-quality crystal of **4**@P³ and a crystal of **3**@P³ of much poorer quality were selected and compared, as shown in Supplementary Fig. 17. Both crystals bent under infrared light, which indicates that the crystal quality has a comparatively smaller effect, although the quantification of this effect is not straightforward.

### Preparation of hybrid organic crystal arrays
Some of the future applications of flexible crystals are based on sensing in two dimensions, which requires an ordered array of individual standing bendable crystals. In order to explore this direction of possible application, we have prepared two-dimensional arrays of the

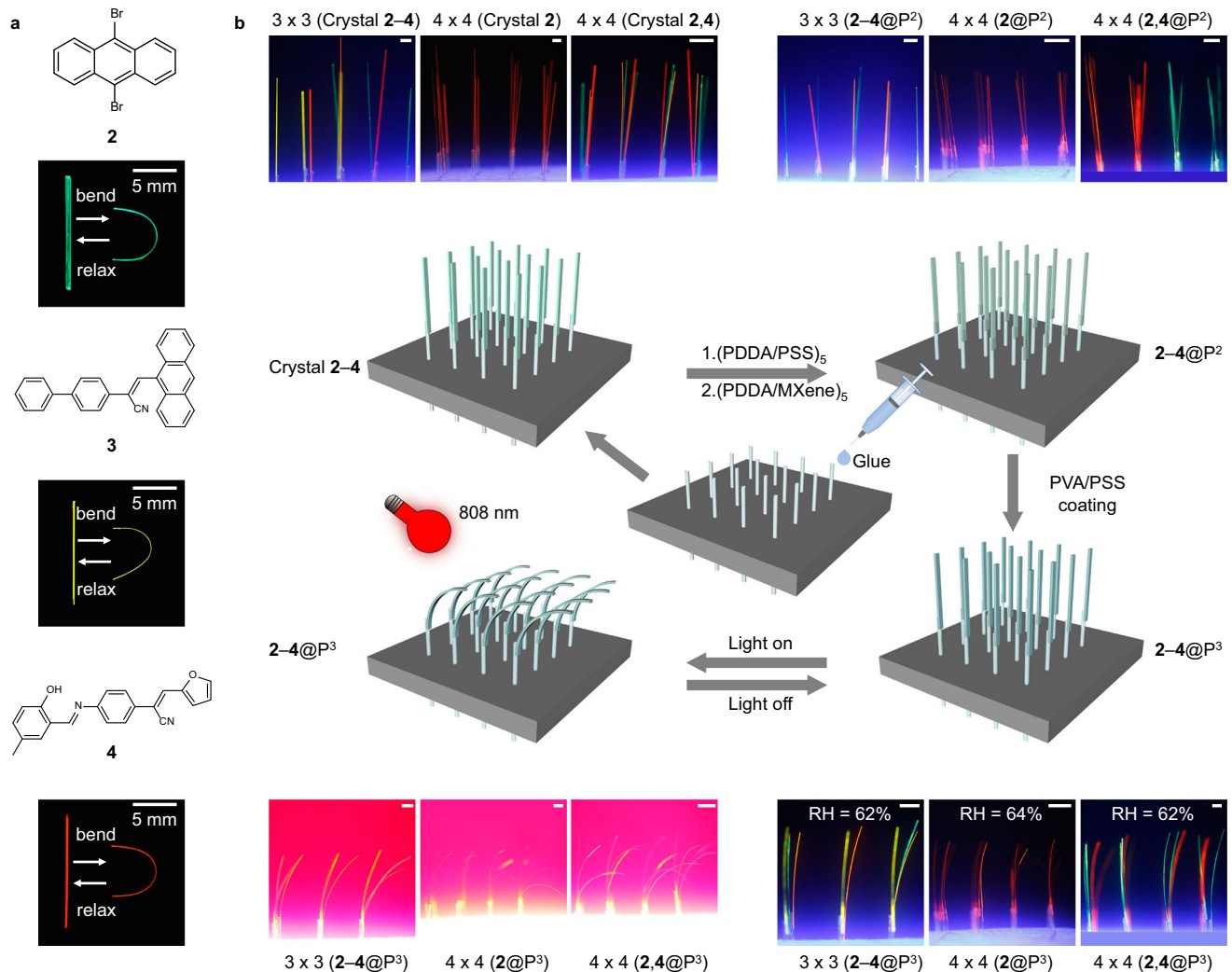

**Fig. 2 | Preparation of hybrid organic crystal arrays. a** The chemical structures of crystals **2–4** and photos (recorded under weak UV light and against a black background for clarity, green is crystal **2**, yellow is crystal **3**, and red is crystal **4**) showing their mechanically induced bending. **b** Process for preparation of hybrid organic crystal arrays (kept at relative humidity RH = 62%, 64%) and their collective bending induced by illumination with infrared light (250 W). Photographs of the arrays are provided next to the illustrations. The blue and red hues are due to reflection of light from the background. All scale bars are 2 mm.

hybrid organic crystals that were bent collectively with infrared light. As shown in Fig. 2b, three different arrays were prepared by gluing crystals to capillaries embedded in a styrofoam base, including $3 \times 3$ arrays of **2–4**@P³, $4 \times 4$ arrays of **4**@P³, and $4 \times 4$ arrays of **2,4**@P³. These crystalline arrays, kept at relative humidity RH = 62%, 64%, were collectively bent in the same direction when exposed to infrared light (250 W). The bending is a result of the absorption of the infrared light by the MXene, which leads to the thermal contraction of the PVA/PSS layer on the crystal's surface. When the irradiation with infrared light is terminated, the hybrid crystals return to their initial, straight shape. To realize a multidimensional bending of these photosensitive arrays, the bending direction of the hybrid organic crystals was controlled. As shown in Fig. 3a, the position of the infrared lamp (184 mW) was fixed, and the capillary glass tube below the styrofoam base was manually rotated (Fig. 2b). As the crystal glued at the tip of the capillary glass tube rotated, it could be bent in different directions. As shown in Fig. 3b–d (Supplementary Movie 3), the $3 \times 3$ array of **2–4**@P³ shows three types of collective bending under infrared light (250 W). Similarly, Fig. 3e–j shows the optomechanical motion of $4 \times 4$ arrays of **4**@P³ and **2,4**@P³. Hybrid crystals with organic crystals having a different size, as well as with organic crystals of the same size but having different thicknesses of the MXene layer were also prepared to

examine the effect of crystal size and MXene thickness on the performance of the 2D array. As shown in Supplementary Fig. 18, four samples of **4**@P³ with different sizes were bent to different angles under identical experimental conditions. Similarly, two hybrid crystals having one or five layers of the MXene, **2**@PVA/PSS@(PDDA/MXene)₁@PDDA/PSS and **2**@PVA/PSS@(PDDA/MXene)₅@PDDA/PSS, shown in Supplementary Fig. 19, bent to a different degree. These results confirmed that both the size of the organic crystal and the thickness of the MXene layer determine the performance of the 2D arrays. The capability of multidirectional motion of the crystal arrays expands the range of their possible applications in two-dimensional optical transmission and detection.

## Durability and sensitivity of hybrid organic crystals

The mechanical robustness, cyclability in operation over prolonged usage without fatigue, and response sensitivity are some of the main prerequisites for future applications of dynamic crystals in devices such as flexible electronic devices. The bending using light provides spatial control over the excitation, which could, in turn, be used to control the shape. A demonstration of this concept is illustrated in Fig. 4a, where four different positions on the same hybrid crystal of **3,4**@P³ were selected for irradiation (184 mW). As shown in

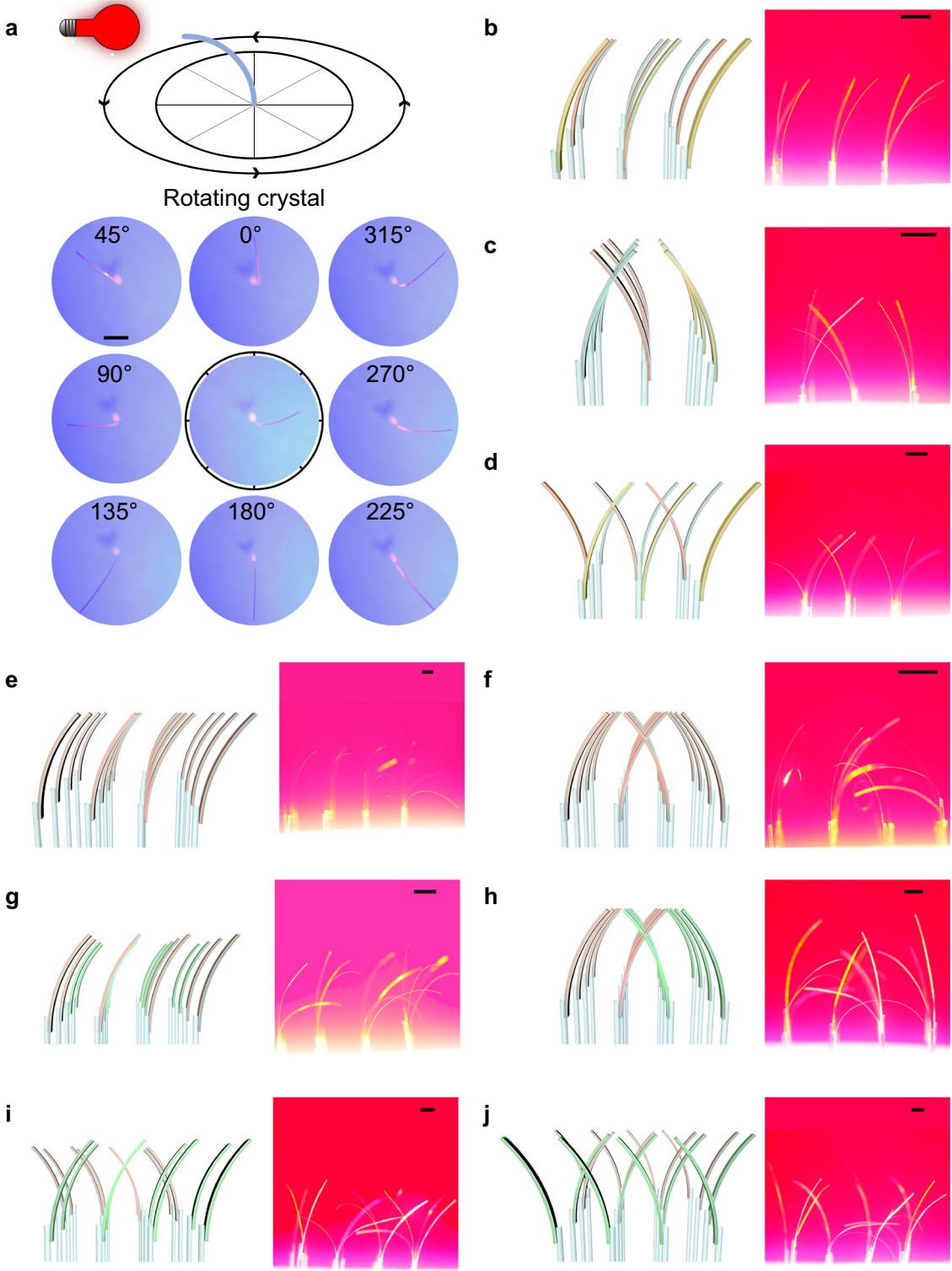

**Fig. 3 | Construction of different types of hybrid organic crystal arrays.**
**a** Photograph of hybrid organic crystal array achieving 360° bending under infrared illumination by rotating a capillary glass tube. **b–j** Schematic diagram and photographs of different types of hybrid organic crystal arrays: **b–d** 3 × 3 arrays of **2–4**@P³, **e–f** 4 × 4 arrays of **4**@P³, and **g–j** 4 × 4 arrays of **2,4**@P³. All scale bars are 2 mm.

Supplementary Fig. 20, the hybrid organic crystals can be bent in both directions by changing the deposition method that had been used to prepare the PVA/PSS film. The shape of the crystal is determined by the location of bending. Moreover, the degree of bending can also be varied by the degree of heating due to light absorption. As shown in Fig. 4b, the bending of the crystals **3**@P³ and **4**@P³ increases with the increasing power of the infrared radiation due to an increase in temperature, mainly of PDDA/MXene (Fig. 1f). **4**@P³ has an inflection point in the bending angle with power at 576 mW, probably due to the fact that the horizontally incident infrared light has access to surfaces of the crystal after it bends over 90°.

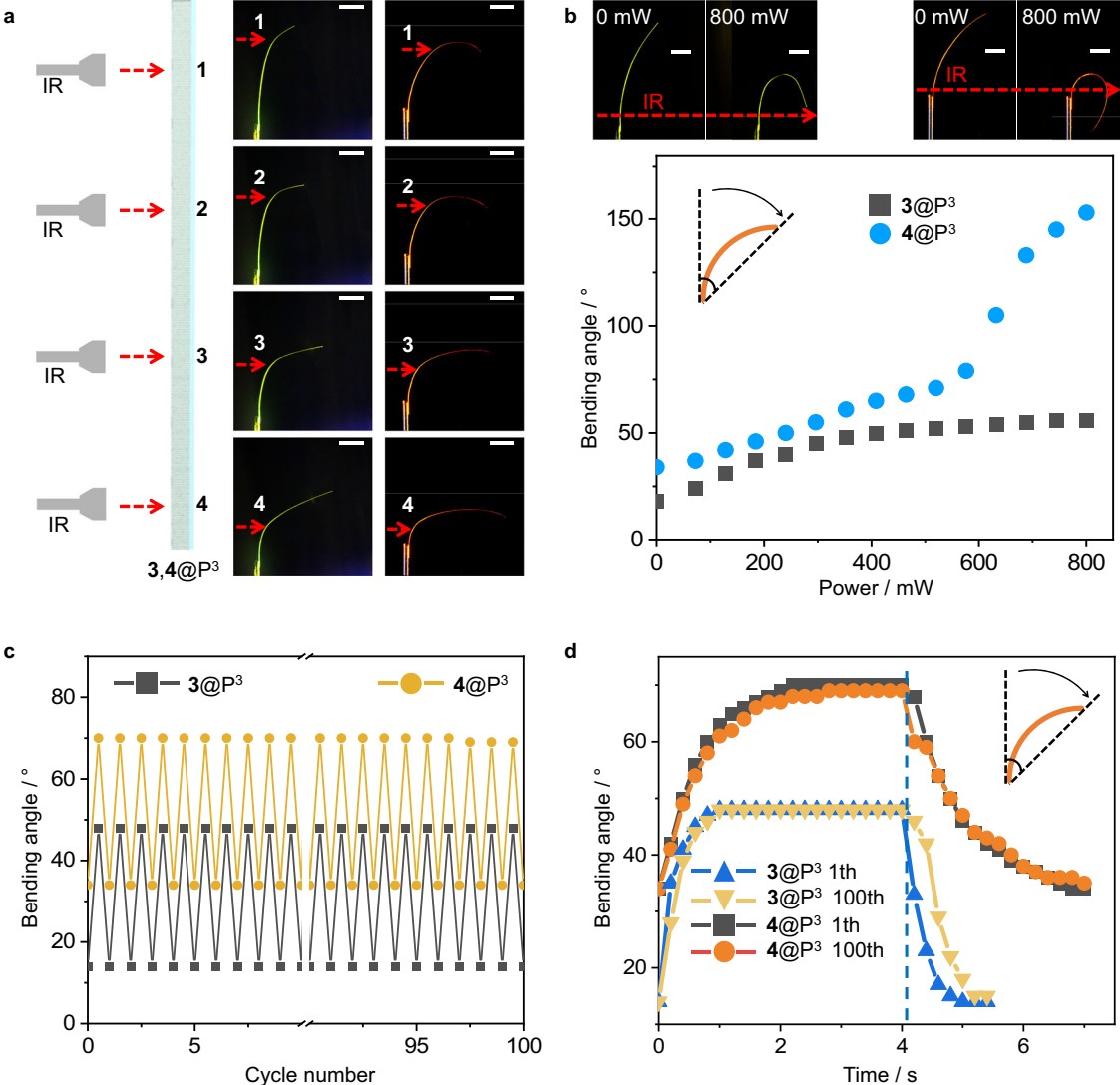

**Fig. 4 | Assessment of the performance of the hybrid organic crystals. a** Bending of **3,4**@P³ induced by irradiation at different positions (1–4) of irradiation with infrared light (184 mW). The red dashed line represents the position excited with infrared light. **b** Photographs showing the degree of bending of **3,4**@P³ (defined in the inset) at different powers of infrared light. The red dashed line represents the position excited with infrared light. **c** Testing of reproducibility of deformation of **3,4**@P³ in cycling mode. **d** Dependence of the bending angle of **3,4**@P³ as a function of time. The dashed blue line separates the light regime (left) and the dark regime, i.e., absence of light (right). All scale bars are 2 mm. Source data are provided as a Source Data file.

Additional tests of the robustness of the hybrid actuators were carried out to characterize the reproducibility of the degree of deformation and sensitivity of **3,4**@P³ under irradiation with infrared light. A light-sensitive element of **3,4**@P³ was exposed to infrared radiation (184 mW) and sunlight (Supplementary Movies 4 and 5). After 100 cycles, the maximum curvature of the bent state remained nearly constant (Fig. 4c). We tentatively attribute the slight decrease to gradual aging of the polymer coating caused by heating. The response rates of the hybrid crystal were estimated during the first cycle and the 100th cycle (Fig. 4d, Supplementary Movies 6–9). The flexing and recovery time of **3,4**@P³ remained nearly constant, at about 1 s and 1 s for **3**@P³, and 2 s and 2.5 s for **4**@P³, respectively. Moreover, fatigue and cycling tests were performed on **3**@P³, and the results confirmed that the performance of the hybrid crystals was retained even after 1000 cycles (Supplementary Fig. 21, Supplementary Movie 10). However, both the rate of bending and the rate of recovery decreased slightly due to a decrease in maximum curvature and hence a decrease in the reverse bending moment.

Overall, we conclude that the performance of these simple light-sensitive devices is sufficiently stable, and this result favors this material as a candidate for practical applications such as optical signal transmission. In addition, as shown in Supplementary Movie 11, we established that the actuation of the hybrid organic crystal bending can be precisely controlled by infrared light at distances of over 3 m (Supplementary Fig. 22). This result further highlights the prospects for long-range sensing, triggering, actuation, or other remote real-world applications of these materials.

## Hybrid organic crystals as optical waveguides

As mentioned above, organic crystals have the advantages of relatively low optical losses and long-range ordered structures. One of the properties that are now actively being explored with slender organic crystals is their optical waveguiding capability, occasionally combined with mechanical flexibility observed with some of these crystals. Recently, similar flexible organic crystals have been shown to hold potential for optical transduction at low temperatures[43].

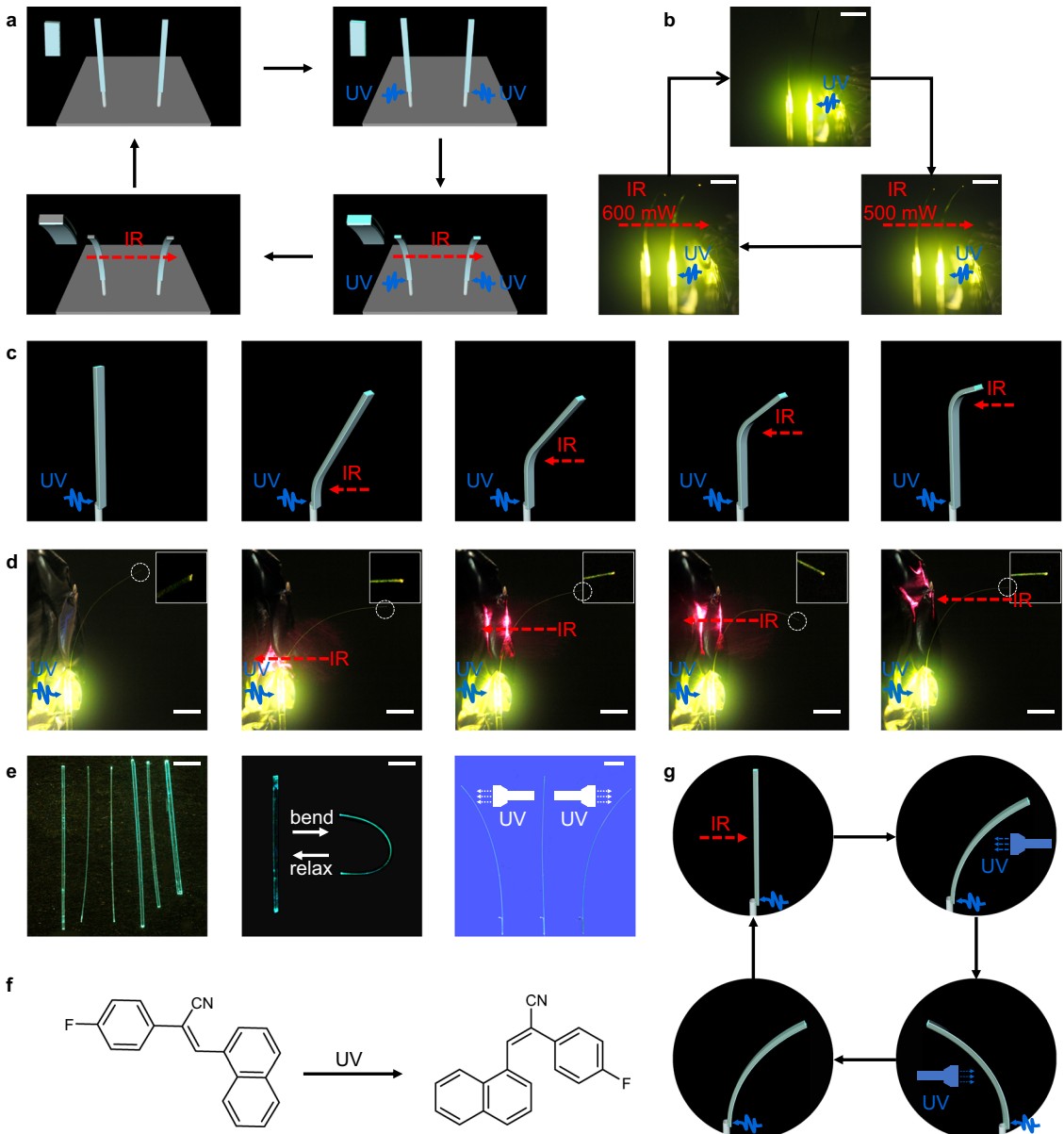

**Fig. 5 | Optical waveguiding properties of the hybrid organic crystals. a** Diagram of infrared light-driven hybrid organic crystal array for optical signal transmission. The top left image is a zoomed-in representation of the crystal tip. **b** Photographs of a hybrid organic crystal array for optical signal transmission. **c** A schematic showing the dependence of the optical output point of a hybrid organic crystal on the excitation position. **d** Photographs showing the change in output of the optical signal of **3@P³** with the position of excitation with infrared light. The insets show 10-fold magnified images of the crystal tip. The broken line circles indicate the position of the optical signal output. **e** Photographs of a crystal of **5** before (left) and after mechanically induced bending (middle) and bending of crystals under UV irradiation (right). **f** Isomerization of **5** exposed to UV light. **g** A schematic showing the concept of an optical waveguide of crystal of **5** controlled by infrared and ultraviolet light. The red dashed arrow indicates the direction of the infrared light, and the blue arrows indicate the direction of the ultraviolet light. Scale bar: 2 mm.

Flexible organic crystalline waveguides for both visible and near-infrared (telecom) ranges of the spectrum have been reported[44–46]. The light output of hybrid organic crystals can be precisely controlled by using a magnetic field or changes in aerial humidity[23,24]. Light-driven bending has also been extensively studied, especially in view of advantages such as remote, long-distance control[47–49]. Organic crystals have been already established as active optical waveguides[9,21,50]. The principles of this property are rooted in their optical transparency and difference in refractive index with air; when one end of the crystal is excited by ultraviolet light, the emitted light is subsequently reflected and refracted continuously inside the crystal and is eventually transmitted to its other end[51].

Figure 5a shows an infrared light-driven array of hybrid organic crystals for optical signal transmission. In Fig. 5b, one end

of the hybrid organic crystal was affixed and excited by a 365 nm laser, and the optical output point at its other end was controlled by infrared radiation. This is accomplished by controlling the degree of bending by changing the point of excitation. As shown in Fig. 5c, d, **3@P³** was excited by a 365 nm laser at its fixed end. The light output at the other end changes with the position of the point of exposure to an 808 nm light (184 mW). **3@P³** was selected to investigate the variation in the bending angle of the hybrid crystal over time under long-term irradiation with infrared light. As shown in Supplementary Movie 12, the bending angle of the hybrid crystal remains almost unchanged up to at least 20 min of exposure to infrared light. Supplementary Movie 14 demonstrates that **3@P³** transmits optical signals for 20 min without visible changes in the direction of the transmission, confirming the

**Fig. 6 | Preparation of compounds 3–5. a** Synthesis of compound **3**. 4-Biphenylacetonitrile and 9-anthraldehyde were added to methanol, followed by NaOH, and stirred for 4 h at room temperature. The resulting crude product was purified by column chromatography using dichloromethane (DCM) as an eluent to obtain compound **3**. **b** Synthesis of compound **4**. 2-(4-Aminophenyl)−3-(furan-2-yl)acrylonitrile and 2-hydroxy-5-methylbenzaldehyde were dissolved in ethanol. After refluxing for 6 h, the resultant mixture was cooled down to room temperature, filtered, and washed with ethanol. The crude product was purified by column chromatography using DCM and petroleum ether as the eluent to produce compound **4**. **c** Synthesis of compound **5**. 4-Fluorophenylacetonitrile and 1-naphthaldehyde were added to methanol, followed by NaOH, and stirred for 1 h at room temperature. The resulting crude product was purified by column chromatography using DCM as eluent to obtain compound **5**.

capability of the hybrid crystal to transmit optical signals in a specific direction over a prolonged period.

Having the infrared-control over the shape of the crystals at hand, hybrid organic crystals that act as optical waveguides and can be controlled by two stimuli, infrared and ultraviolet light, were also prepared. A centimeter-long slender elastic crystal of the compound (*E*)-2-(4-fluorophenyl)−3-(naphthalen-1-yl)acrylonitrile (hereafter referred to as **5**; Fig. 5e, f) that has been reported earlier[24] was selected for the purpose. The crystal is elastic and has Young's modulus of 2.49 GPa. It can be bent repeatedly without breaking (Fig. 5e) (Supplementary Fig. 23). Crystal of **5** can also be bent by irradiation with UV light due to configurational isomerization, and this process was confirmed by NMR spectroscopy (Fig. 5f; Supplementary Fig. 24). In the experiment shown in Fig. 5g and Supplementary Movie 13, **5**@P³ is excited by a 355 nm laser at its fixed end, and the optical output point at the other end is controlled by infrared or ultraviolet light. As shown in Supplementary Movie 14, **3**@P³ transmits the optical signal for 20 min without visible changes in the direction of optical transmission, confirming the capability of the hybrid crystal to transmit optical signals in a specific direction over prolonged periods of time. To confirm that the hybrid organic crystal was conductive to light, the optical waveguiding capability of **3,5**@P³ in the straight and bent states was tested by using a standard method[52]. The distance-dependent emission spectra were obtained by irradiating different positions of **3,5**@P³ by using a 355 nm laser (10 Hz, 10 ns), collecting the emission spectra at the other end of the crystal and fitting the data (Supplementary Fig. 25a–d). Expectedly, the emission intensity of the tip gradually decreased as the distance from the irradiation position to the tip increased due to the increased loss of emitted light with distance (Supplementary Fig. 25e–h). The optical loss coefficients were found to be 0.15461 dB mm⁻¹ and 0.15562 dB mm⁻¹ for the straight and bent state of **3**@P³, and 0.12091 dB mm⁻¹ and 0.12689 dB mm⁻¹ for **5**@P³ (Supplementary Fig. 25i–l). Moreover, the optical loss of the hybrid crystal was measured after 100 bending cycles and long exposure time. As shown in Supplementary Fig. 26, the optical loss of **3**@P³ at 0, 50, and 100-fold bending was 0.16373, 0.16988, and 0.17649 dB mm⁻¹, respectively. The optical loss after irradiation of 0, 30, and 60 min was 0.15600, 0.16238, and 0.17458 dB mm⁻¹, respectively (Supplementary Fig. 27). These results clearly confirmed that the light transmission through the hybrid organic crystal can be controlled by both infrared and ultraviolet light.

In summary, here we report a family of hybrid organic crystalline materials whose deformation is driven and can be precisely controlled by infrared light. The hybrid materials display favorable properties which combine mechanical flexibility brought about by the mechanical compliance of the organic crystals, on the one hand, and thermal sensitivity, governed by a layer of MXene on their surface, on the other. By simply adjusting the infrared light, these hybrid organic crystals can be bent to an arbitrary extent and at a desired position along their length. This deformation can be induced not only with individual crystals but also with a collection of hybrid crystals arranged in a regular two-dimensional array. The hybrid materials described here have the advantages of high sensitivity, high and controllable degree of deformation, and durability over prolonged actuation, while the bending point of each crystal can also be controlled by changing the position of excitation with infrared light. As a proof-of-concept of the immense opportunities that this approach opens for dynamic materials, we demonstrate that infrared light-driven flexible organic crystal optical waveguides can be constructed from these materials, including ordered arrays of such optical waveguides. Since photothermal bending has some advantages over other modes of excitation, such as the possibility to control the point of bending by choosing the position of excitation and to deform the crystal remotely by irradiation over long distances, adding the infrared light to the palette of available excitation stimuli such as UV or visible light, humidity, and magnetic field expands significantly the prospects for construction of flexible optical and electronic devices based on organic crystals.

## Methods
### Materials
All solvents and starting materials for the synthesis were purchased from commercial sources and were used as received. PDDA, $M_w$

200,000–350,000 g mol⁻¹ (99%), PSS, $M_w$ 70,000 g mol⁻¹ (99%), and PVA, $M_w$ 105,000 g mol⁻¹ (99%), were purchased from Energy Chemical. The PDDA and PSS solutions in water were at a concentration of 1.0 mg mL⁻¹. To prepare the 5% PVA aqueous solution, after mixing PVA granules with pure water in a mass ratio PVA : H₂O = 5 : 95, the suspension was first stirred mechanically at room temperature for 2 h, and then stirred at 95 °C in a water bath for 2 h. A clear solution of 5% PVA was obtained. To prepare the PVA/PSS mixture, 5% PVA solution was mixed with 30% PSS solution in a mass ratio PSS : PVA = 1 : 2 and stirred overnight by using a magnetic stirrer. The ¹H NMR spectra were recorded on an Agar Scientifica 400 MHz instrument. Scanning electron microscopy (SEM) images were obtained on the FEI Quanta 450 operated at 5–10 kV. The emission spectra were recorded on a Maya2000 Pro CCD spectrometer. For the optical waveguide tests, the crystals were irradiated by the third harmonic (355 nm) of a Nd:YAG (yttrium-aluminum garnet) laser at a repetition rate of 10 Hz and a pulse duration of about 10 ns. The energy of the laser was adjusted by using calibrated neutral density filters. The beam was focused on a stripe whose shape was adjusted to 3.3 × 0.6 mm² by using a cylindrical lens and a slit. The atomistic data were obtained on the BRUKER ICON-XR. Differential scanning calorimetric (DSC) measurements were performed by using a TA Instruments DSC Q20 calorimeter.

### Preparation of compounds 1–5 and their crystals

Compounds **1** (98%) and **2** (98%) were purchased from Energy Chemical. The compounds **3–5** were synthesized following established procedures (Fig. 6)[24,42]. 4-Biphenylacetonitrile (98%) (1.93 g, 10 mmol) and 9-anthraldehyde (97%) (2.06 g, 10 mmol) were added to ethanol (50 mL). NaOH (0.40 g, 10 mmol) was then added, and the mixture was stirred for 4 h at room temperature. The mixture was filtered to have a yellow solid, which was dissolved in dichloromethane and washed with brine. After drying over Na₂SO₄, the solvent was removed by vacuum roto-evaporation. The resulting crude product was purified by column chromatography using dichloromethane as an eluent and compound **3** (3.52 g, 88%) was obtained as a yellow powder (Fig. 6a). 2-(4-Aminophenyl)−3-(furan-2-yl)acrylonitrile (98%) (2.10 g, 10 mmol) and 2-hydroxy-5-methylbenzaldehyde (97%) (1.36 g, 10 mmol) were dissolved in ethanol (50 mL). After refluxing for 6 h, the resultant mixture was cooled down to room temperature and was filtered and washed with ethanol. The crude product was purified by column chromatography using dichloromethane and petroleum ether (v/v = 4:1) as the eluent to obtain compound **4** as an orange-red solid (2.88 g, 83% yield) (Fig. 6b). 4-Fluorophenylacetonitrile (98%) (1.36 g, 10 mmol) and 1-naphthaldehyde (97%) (1.56 g, 10 mmol) were added to methanol (50 mL). NaOH (0.40 g, 10 mmol) was added and the mixture was stirred for 1 h at room temperature. The reaction mixture was filtered to give a green solid, which was dissolved in dichloromethane and washed with brine. After drying over Na₂SO₄, the solvent was removed by vacuum roto-evaporation. The resulting crude product was purified by column chromatography using dichloromethane as eluent to obtain compound **5** (2.33 g, 80%) as a green powder (Fig. 6c). To prepare the samples, dichloromethane solutions of **1–5** were placed in test tubes. Approximately triple volume of ethanol was then added along the walls of the tube without disturbing the surface of the solution and the solutions were left undisturbed for diffusion to occur. Needle-shaped crystals of the compounds **1–5** were obtained after 1–2 weeks at room temperature.

### Preparation of MXene

To prepare the $Ti_3C_2T_x$ MXene suspension, $Ti_3AlC_2$ powder was purchased from Energy Chemical; 2.0 g $Ti_3AlC_2$ powder was mixed with 2.0 g LiF and 40 mL 9.0 M HCl, and stirred at 35 °C for 24 h. The mixture was washed by centrifugation at 3500 rpm several times until the pH of the supernatant was 7. Then the sediment of MXene was collected and mixed with 40 mL of deionized water. The mixture was sonicated under nitrogen for 1 h and then centrifuged at 3500 rpm for 1 h to obtain the MXene suspension. The MXene powder was prepared by freeze-drying of the MXene suspension (1 mg mL⁻¹).

### Fabrication of the hybrid crystals 1–5@P³

The long crystals were immersed in 1 mg mL⁻¹ solution of PDDA for 20 min, followed by 1 min rinse with distilled water. Then, the crystals were immersed in 1 mg mL⁻¹ solution of PSS for 20 min, and rinsed with distilled water for 1 min. The coated crystals were obtained by repeating the above steps. Subsequently, the above polymer-coated crystals were immersed in a solution of PDDA for 20 min, followed by 1 min rinse with distilled water. The crystals were then immersed in 1 mg mL⁻¹ of MXene solution for 20 min and rinsed with distilled water for 1 min. Crystals coated with MXene were obtained by repeating the above steps. By using a needle tip, a mixture of PVA, PSS was deposited on one of the crystal's wide faces. As the solvent evaporated at room temperature, a polymer film formed on the surface of the crystal, and **1–5@P³** were obtained.

## Data availability

All data are available from the corresponding authors upon request. Source data are provided with this paper.

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

## Acknowledgements

This work was supported by the National Natural Science Foundation of China (52173164, 51972228), the Natural Science Foundation of Jilin Province (20230101038JC), and a fund from New York University Abu Dhabi. This material is based upon works supported by Tamkeen under NYUAD RRC Grant No. CG011.

## Author contributions

X.Y., L.Lan, L.Li, J.Y., and X.L. performed the experiments. Y.T., Q.Y., P.N., and H.Z. supervised the experiments. H.Z. and P. N. conceived the project.

## Competing interests

The authors declare no competing interests.
