## [Peer Review File · Nature Communications]

Reviewer comments, first round review

Reviewer #1 (Remarks to the Author):

In this manuscript, the authors fabricated an infrared light-driven actuator based on MXene-polymer hybrid with an organic crystal. The fabricated waveguiding fiber could bend more significantly than other waveguiding fibers due to the photothermal effects of MXene. Additionally, the response speed and robustness were reasonably good. However, I believe that the concept and idea of the paper may not meet the high standards of Nature Communications. Bendable waveguiding organic crystals have been frequently reported in other journals, and using photothermal effect is not a novel method. Also, it is difficult to find the advantages of using photothermal effects over other methods, including light, humidity, or magnetic field. Furthermore, the mechanism of determining the bending direction seems uncertain. I hope the authors reinforce their opinion on the advantages of photothermal methods and provide a clear explanation on setting the direction of bending on the fiber.

1. The authors used the photothermal effect to bend flexible organic crystals. However, there is a lack of a unique advantage of using photothermal effects. Please provide a unique advantage of using photothermal effects on flexible organic crystals.

2. In Figure 2B, the authors displayed RH information in the figure. However, I could not find any related text in the manuscript.

3. What happens to organic crystals at different humidities, such as RH 20% or 80%?

4. In Figure 3, the authors showed the rotation of crystals by changing the position of the glass tube. However, there is a lack of information on how the rotation was possible. Additionally, I believe that, to prove the controllability of the crystals, the authors should fix the glass tube and change the incident light.

5. In Figure 4B, in the case of sample #4, there is an abrupt change in the bending angle. Why does this phenomenon happen?

6. In Figure 4D, the authors showed the mechanical stability results of the crystal by showing the bending angle of the crystals. However, as the authors emphasized the waveguiding performance of the organic crystals, it could increase readers' understanding by showing optical loss data after 100 folds. Additionally, what happens to optical loss after a long period of input light? The period of the light pulse is missing in the manuscript.

7. In Figures 5C and D, it is hard to distinguish the position of the crystal ends and input light. Please provide additional position information for the crystal ends and input, as the authors did in Figure 4B.

Reviewer #2 (Remarks to the Author):

In this article, the authors explored the flexible elastic crystals for optical waveguides. The needle-shaped crystals were converted into hybrid materials by coating them with layers of polymers and MXene. This hybrid material is explored for IR light-driven flexible material for optical waveguides, where the hybrid crystal array responds to the dual stimuli differently (IR and UV light). This is indeed one step ahead in the progress of crystal engineering and its application in various optoelectronic devices.

The authors should address the following two comments before publication.

1. From the discussion and methodology, it is not very clear how the hybrid crystals are created exactly. More details procedures are required for the coating experiments.

2. There should be more discussion on the mechanisms of different observation such as bending and waveguide.

Reviewer #3 (Remarks to the Author):

This is an interesting piece of work. The application of MXene-based materials in the field of

flexible optical waveguide is a novel subject. This work provides a simple and effective way to realize the flexibility of optical waveguide on organic crystals, and this method is applicable to a variety of materials. Moreover, 2D array with high sensitivity, high and controllable degree of deformation and durability over prolonged actuation is also carried out, while the bending point of each crystal can be controlled by changing the infrared light position. Therefore I recommend this article can be considered for publication after the authors have addressed the following concerns:

- 1)The bending of organic crystals caused by thermal effect should be further analyzed theoretically. Does it involve physical changes such as phase transition or chemical changes?
- 2)Does the sample size or deposition thickness affect the performance of the 2D array?
- 3)The authors state that "the organic crystals have the advantages of having defect-free structures", more characterization is needed to support this claim.
- 4)When it comes to fatigue and cycling, 100 times bending test is not enough. More cycles should be shown.
- 5)There are some typos and mistakes: the font should be uniform in Line 51, redundant words in Line 109, etc.

Reviewer #4 (Remarks to the Author):

The manuscript describes the realization of rod-shaped organic crystals that exhibit a deformation response to IR light. This photoresponse is enabled by coating the crystals in a multilayer structure containing MXenes that absorb the IR irradiation. This absorption produces a local thermal response that triggers the actuation mechanism of the organic crystals. The work demonstrates this effect using a few different chemistries of organic crystals. The manuscript also shows that the location of the deformation can be controlled through the location of the illumination.

The manuscript is convincing in its conclusion and has a clear message. I have no doubt that the crystals exhibit the deformation described by the authors. However, I have a difficult time judging the novelty and impact of the work. The manuscript is quite applied without much decision of scientific mechanisms, material design considerations, or quantitative analysis. I believe the authors should address a few specific points:

1. The materials should be described in much more detail. I do not believe referring to the crystals as "1" or "2" is appropriate. What are the chemical names or formula of the compounds?

2. Do the authors have any materials characterization related to the actual materials? Have more than one of the same composition of crystal been tested? Is there any relationship between structure/quality of the crystals and the photo-mechanical response?

3. Aspects of Figure 1 are unclear. 1d shows "photographs" of the samples. What is the difference between images 0-5 in that panel? Are these the same samples as shown in the SEM images in Fig 1b? Is the AFM data in Fig 1c free from tip artifacts? The right image has streaky features that all look quite similar; do these persist when the scan direction (or sample) is rotated 90 degrees?

4. The authors state that the organic crystals have "defect-free structures". No characterization is provided to support this. Just from a basic thermodynamic standpoint, no materials are defect-free and such statements should be removed. In Fig. S2, macroscopic defects are clearly visible in the crystals.

5. The authors should provide details of how they synthesized the MXene. The statement in the supplemental that "The MXene was synthesized by using literature methods" is insufficient.

Response to reviewers' comments

Response to the comments from Reviewer #1:

General comments: *In this manuscript, the authors fabricated an infrared light-driven actuator based on MXene-polymer hybrid with an organic crystal. The fabricated waveguiding fiber could bend more significantly than other waveguiding fibers due to the photothermal effects of MXene. Additionally, the response speed and robustness were reasonably good. However, I believe that the concept and idea of the paper may not meet the high standards of Nature Communications. Bendable waveguiding organic crystals have been frequently reported in other journals, and using photothermal effect is not a novel method. Also, it is difficult to find the advantages of using photothermal effects over other methods, including light, humidity, or magnetic field. Furthermore, the mechanism of determining the bending direction seems uncertain. I hope the authors reinforce their opinion on the advantages of photothermal methods and provide a clear explanation on setting the direction of bending on the fiber.*

Response: We thank the Reviewer for the insightful comments. We acknowledge the Reviewers' concerns and the comments related to the advantages of photothermal methods. In our detailed response to their comments below we provide explanation on the importance and advantages of using the photothermal effect and the reason for setting of the direction of bending on the fiber.

Comment: 1. *The authors used the photothermal effect to bend flexible organic crystals. However, there is a lack of a unique advantage of using photothermal effects. Please provide a unique advantage of using photothermal effects on flexible organic crystals.*

Response: We thank the Reviewer for pointing this out. The photothermal effect has been studied extensively in case of photoactive polymers, such as, for example, liquid-crystalline elastomers, where it has been used to induce rapid deformation. Recently, it was also reported for single crystals or organic molecules. Here, for the first time, we report the effect of photothermal heating on *hybrid* single crystals. There are several advantages of using the photothermal effect to actuate crystals, one of which is the potential of using light to actuate even crystals which are not photochemically reactive. This is significant advantage which expands the application to any organic crystal. The second advantage is the high speed of deformation which can be achieved by using the thermal effects. The third advantage is the opportunity for remote control of the shape of the crystal, similar to other photoinduced mechanical effects, however in case of photothermal effects this can be performed over very long distances. Finally, the fourth advantage is that by using photothermal effect, one can bend the crystal at specific, predetermined locations, that is, it is possible to have a spatial control over the crystal's deformation. To highlight these points, we modified the abstract, the introduction, and some of the main text of the manuscript.

The following text was added to the introduction section: "The photothermal actuation of the hybrid crystals reported here comes with multiple advantages over other methods that have been used for crystal deformation. Since the absorber is the MXene, this approach does not require absorption of radiation by the crystal itself, which circumvents the necessity for the crystal to absorb light (i.e. be photoreactive). Second, both high speeds and high deformations can be achieved, and the crystal can be actuated by localized thermal excitation at a predetermined location. Finally, we demonstrate that hybrid crystals can be precisely actuated

by excitation over very long distances (> 3 meters), which brings an added value to the potentials for real-world applications of dynamic responsiveness of organic crystals, where they could be used as receivers for remote sensing, triggering, or actuation.”

The following sentences were added to the abstract (one sentence was deleted to shorten the abstract and keep it within the limits): “Photothermal actuation of the organic-MXene crystals circumvents the limiting requirement for photoreactive groups in the organic crystal, and can be applied to any flexible crystal. The method provides high deformation speeds with precisely controlled, localized deformation.”

“We also demonstrate that excitation can be achieved over very long distances (> 3 meters), which places these materials into the realm of long-distance receivers for remote sensing, triggering, or actuation.”

The following text has been revised at the end of the conclusions section:

Original text: “Adding the IR light to the palette of available excitation stimuli such as light, humidity and magnetic field expands significantly the prospects for construction of flexible optical and electronic devices based on organic crystals.”

Revised text: “Since photothermal bending has unique advantages over other modes of excitation, such as the possibility to control the point of bending by choosing the position of excitation and to deform the crystal remotely by irradiation over long distances, adding the IR light to the palette of available excitation stimuli such as light, humidity, and magnetic field expands significantly the prospects for construction of flexible optical and electronic devices based on organic crystals.”

Some of the above advantages are clearly illustrated in our manuscript. For example, as shown in Figure 4a and 5d, flexible organic crystals can be bent at different positions along the crystal, and this can be achieved by controlling the position of irradiation with infrared light. By using the photothermal effect, bending of flexible organic crystal can also be achieved by irradiation over long distances. To emphasize this latter point, Supplementary Movie 9 has been modified, where we show an excitation of the crystal over a distance of 3 meters.

Accordingly, the following text have been revised:

Original text: “In addition, we established that the actuation of the hybrid organic crystal bending can be controlled by infrared light over long distances (>2 m), as shown in the Supplementary Movie 9, a result which highlights the prospects for long-range sensing or other remote applications of these materials”

Revised text: “In addition, as shown in the Supplementary Movie 10, we established that the actuation of the hybrid organic crystal bending can be precisely controlled by infrared light at distances of over 3 meters (Supplementary Figure 20). This result further highlights the prospects for long-range sensing, triggering, actuation, or other remote real-world applications of these materials.”

New Supplementary Figure 20. Bending of hybrid crystal over long distances. The image on the right shows a hybrid crystal that has been bent by exposure to IR light where the light source is more than 3 meters away from the crystal.

Comment: 2. *In Figure 2B, the authors displayed RH information in the figure. However, I could not find any related text in the manuscript.*

Response: We thank the Reviewer for bringing this important point to our attention. The information on the RH has been added to the relevant text in the manuscript.

The following text has been revised on page 6, line 161 in the revised manuscript:

Original text: “(b) Process for preparation of hybrid organic crystal arrays and their collective bending induced by illumination with infrared light (250 W).”

Revised text: “(b) Process for preparation of hybrid organic crystal arrays (kept at relative humidity $RH = 62\%$, 64%) and their collective bending induced by illumination with infrared light (250 W).”

The following text has been revised on page 6, line 170 in the revised manuscript:

Original text: “These crystalline arrays can be collectively bent in the same direction when exposed to IR light (250 W).”

Revised text: “These crystalline arrays, which were kept at relative humidity $RH = 62\%$, 64% , were collectively bent in the same direction when exposed to IR light (250 W).”

Comment: 3. *What happens to organic crystals at different humidities, such as $RH 20\%$ or 80% ?*

Response: We are thankful to the Reviewer for this important question. MXene-Polymer hybrid crystal bending is a result of the infrared light absorption by the MXene, which leads to thermal contraction of the PVA/PSS layer on the crystal surface. The PVA/PSS layer functions as the element which causes differential strain that results in bending moment. PVA is a common hygroscopic polymer with low critical solubility temperature (LCST) that is well-known to undergo reversible swelling by formation of hydrogen bonds. Therefore, the hybrid material is sensitive to humidity, and the bending angle could be changed by changes in humidity.

In order to clarify this point, the following text has been added on page 5, line 134 in the revised manuscript, and a new Supplementary Figure (Figure 6) was also added:

Added text: “In the hybrid crystal, the MXene functions as a photothermal converter, while the PVA/PSS layer functions as one of the two components of a bilayer strip that generates bending moment by expansion or contraction. PVA is a common hygroscopic polymer that has a low critical solubility temperature, and is well known to undergo reversible swelling via hydrogen bond formation (Supplementary Figure 6).⁴³ This brings about the response of the hybrid element to humidity, and the curvature could change with variation in humidity (Supplementary Figure 7).”

New Supplementary Figure 6. Mechanism driving the bending by photothermal effect of the hybrid crystals. The diagram shows swelling or contraction of the polymer layer induced by heating by the MXene induced by infrared light.

New Supplementary Figure 7. Humidity response of the hybrid organic crystals. Photographs showing P³//3 that bends upon exposure to different aerial humidity.

Comment: 4. In Figure 3, the authors showed the rotation of crystals by changing the position of the glass tube. However, there is a lack of information on how the rotation was possible. Additionally, I believe that, to prove the controllability of the crystals, the authors should fix the glass tube and change the incident light.

Response: We are grateful for this comment, which brings the necessity for some clarification of the experiment. We apologize that the details of rotation of the capillary glass tube to change the bending direction of the hybrid crystal were not clearly described in the original manuscript. The relevant details have now been added to the text. In addition, the experiment of fixing the glass tube and changing the incident light, which was suggested by the Reviewer, was also performed. As shown in Figure R1 below, when the direction of the incident IR light (left and right) is changed, the bending direction of the hybrid crystal does not change. The reason for this is that, in this work, the polymer was deposited on only one of the wide faces of the crystal (as described on page 5, line 132). When the MXene is heated by light, the polymer shrinks, which affects only the deposited face, and this leads to differential strain at the interface, resulting in bending of the crystal. However, if the deposition method of the PVA/PSS film is modified, bending of the crystal in both directions can be achieved when the capillary glass tube is fixed. As shown in the Supplementary Figure 18, the hybrid crystal can indeed be bent in both directions by changing the direction of the incident infrared light.

Figure R1. Hybrid crystal bending under IR light irradiation. (a) Photograph of P³//3 bending under illumination with infrared light from right ($RH = 79\%$). (b) Photograph of P³//3 bending under illumination with infrared light from left ($RH = 79\%$).

To clarify this point, the following text has been revised (page 7, line 175):

Original text: “The position of the infrared lamp (184 mW) was fixed, and turning of the capillary glass tube results in multidirectional bending of the hybrid crystals (Figure 3a).”

Revised text: “As shown in Figure 3a, the position of the infrared lamp (184 mW) was fixed, and the capillary glass tube below the styrofoam base was manually rotated (Figure 2b). As the crystal glued at the tip of the capillary glass tube rotated, it could be bent in different directions.”

The following text has been added (page 8, line 205):

Added text: “As shown in Supplementary Figure 18, the hybrid organic crystals can be bent in both directions by changing the deposition method that was used to prepare the PVA/PSS film.”

New Supplementary Figure 18. Bending of hybrid organic crystals in two directions. (a) A schematic describing the PVA/PSS film deposition. (b) Photographs of P³//4 bending in different directions under infrared light irradiation (*RH* = 79%).

Comment: 5. *In Figure 4B, in the case of sample #4, there is an abrupt change in the bending angle. Why does this phenomenon happen?*

Response: We are grateful to the Reviewer's patience with the detailed reading, which really helped us provide better clarification of some of the experimental details. In Figure 4B, the abrupt change in the bending angle for sample #4 could be interpreted as follows. Over the bending process, especially when P³//4 bends beyond 90°, the IR beam could illuminate multiple positions of the crystal P³//4 at the same time, which causes the bending to increase further. This possible and very probable reason for the observation has been explained in the text (page 9, line 210).

Comment: 6. *In Figure 4D, the authors showed the mechanical stability results of the crystal by showing the bending angle of the crystals. However, as the authors emphasized the waveguiding performance of the organic crystals, it could increase readers' understanding by showing optical loss data after 100 folds. Additionally, what happens to optical loss after a long period of input light? The period of the light pulse is missing in the manuscript.*

Response: We thank the Reviewer for the suggestions. We concur with the Reviewer that, indeed, the reader's understanding can be enhanced by showing the optical loss data after 100 folds. In response to this suggestion, the optical loss after a long period of input light and the period of the light pulse have been added to the text, and new Supplementary Figures 24 and 25 were added with this data.

The following text has been revised on page 11, line 270 in the revised manuscript:

Original text: “The distance-dependent emission spectra were obtained by irradiating different positions of P³//3,5 by using a 355 nm laser and the emission spectra were collected at the other end of the crystal and fitted (Supplementary Figure 5a–d).”

Revised text: “The distance-dependent emission spectra were obtained by irradiating different positions of P³//3,5 by using a 355 nm laser (10 Hz, 10 ns) and collecting the emission spectra at the other end of the crystal and fitting the data (Supplementary Figure 23a–d).”

Moreover, the following text has been added on page 11, line 276 in the revised manuscript:

Added text: “Moreover, the optical loss of the hybrid crystal was measured after 100 bending cycles and a long time of exposure. As shown in Supplementary Figure 24, P³//3, the optical loss at 0, 50, 100-fold bending was 0.16373, 0.16988 and 0.17649 dB mm⁻¹ respectively. The optical loss after irradiation of 0, 30 and 60 minutes was 0.15600, 0.16238, and 0.17458 dB mm⁻¹, respectively (Supplementary Figure 25).”

New Supplementary Figure 24. Dependence of the optical loss on the crystal bending cycles. (a–c) Fluorescence spectra were collected at the fixed end of the crystal, while the crystals were excited at different position by 355 nm laser (10 Hz, 10 ns). The difference in position between the fixed end and the excitation position is defined as distance (mm). Panels a, b, and c correspond to optical loss after 0, 50, 100-fold bending, respectively. (d–f) Decay of intensity with distance $I_{\text{tip}}/I_{\text{body}}$. The optical loss coefficient (α) was obtained by a single exponential fitting function $I_{\text{tip}}/I_{\text{body}} = A \exp(-\alpha D)$, where I_{tip} and I_{body} are the fluorescence intensities measured at the fixed end and the excitation position, respectively. A is the optical loss coefficient and D is the distance between the fixed end and the excitation position. The panels show P³//3 after 0 (d), 50 (e), and 100 (f) bending cycles.

New Supplementary Figure 25. Dependence of the optical loss on the duration of excitation. (a–c) Fluorescence spectra were collected at the fixed end of the crystal, while the crystals were excited at different position by a 355 nm laser (10 Hz, 10 ns). The difference in position between the fixed end and the excitation point is defined as distance (mm). Panels a, b, and c correspond to optical loss after excitation of 0 min, 30 min, and 60 min, respectively. (d–f) Decay of intensity with distance I_{tip}/I_{body} . The optical loss coefficient (α) was obtained by a single exponential fitting function $I_{tip}/I_{body} = A \exp(-\alpha D)$, where I_{tip} and I_{body} are the fluorescence intensities measured at the fixed end and the excitation position, respectively. A is the optical loss coefficient and D is position differences between the fixed end and the excitation position. The panels show $P^3/3$ at 0 min (d), at 30 min (e), and at 60 min (f) of irradiation.

Comment: 7. In Figures 5C and D, it is hard to distinguish the position of the crystal ends and input light. Please provide additional position information for the crystal ends and input, as the authors did in Figure 4B.

Response: We thank the Reviewer for pointing out this important detail. Figure 5c and 5d has been modified and the revised figure was included in the manuscript.

New Figure 5. Optical waveguiding properties of the hybrid organic crystals. (a) Diagram of IR light-driven hybrid organic crystal array for optical signal transmission. The top left image is a zoomed-in representation of the crystal tip. (b) Photographs of a hybrid organic crystal array for optical signal transmission. (c) A schematic showing the dependence of the optical output point of a hybrid organic crystal on the excitation position. (d) Photographs showing the change in output of the optical signal of P³//3 with the position of the IR light. The insets show 10-fold magnified images of the crystal tip. The broken line circles indicate the position of the optical signal output. (e). Photographs of a crystal of **5** before (left) and after mechanically induced bending (middle) and UV-light-induced bending (right). (f) Isomerization of **5** exposed to UV light. (g) A schematic showing the concept of an optical waveguide of crystal of **5** controlled by IR and UV light. The red broken line indicates the direction of the IR light, and the blue arrow represents the direction of the UV light.

Response to the comments from Reviewer #2:

General comments: *In this article, the authors explored the flexible elastic crystals for optical waveguides. The needle-shaped crystals were converted into hybrid materials by coating them with layers of polymers and MXene. This hybrid material is explored for IR light-driven flexible material for optical waveguides, where the hybrid crystal array responds to the dual stimuli differently (IR and UV light). This is indeed one step ahead in the progress of crystal engineering and its application in various optoelectronic devices.*

The authors should address the following two comments before publication.

Response: We are grateful to the Reviewer for their constructive and encouraging comments, and the overall positive assessment of our manuscript. Below, we provide a point-by-point response to their comments and suggestions.

Comment: 1. *From the discussion and methodology, it is not very clear how the hybrid crystals are created exactly. More details procedures are required for the coating experiments.*

Response: In response to the Reviewer's comment, additional details on the procedure used for preparation of the hybrid crystals and the coating were provided in the revised version of the manuscript. Specifically, the following text was revised and expanded (page 5, line 123), and a new Supplementary Figure 5 was added to the supplementary materials that illustrates the process of preparation of the hybrid crystals.

Original text: "In Figure 2b, nascent (as-crystallized) crystals of **2–4** were first uniformly coated with a mixture of poly(diallyldimethylammonium) (PDDA) and poly(styrene sulfonate) (PSS) layer of ca. 650 nm thickness. The surfaces of the resulting hybrid crystals, PDDA/PSS//**2–4** (hereafter, P//**2–4**), were then coated with a mixture of PDDA and MXene layer of ca. 200 nm thickness, giving hybrid materials that can be described as PDDA/MXene//PDDA/PSS//**2–4** (for convenience, hereafter referred to as P²//**2–4**). Lastly, by using a needle tip, a 2 μm-thick layer of polyvinyl alcohol/poly(sodium 4-styrenesulfonate) (PVA/PSS) was deposited uniformly and rapidly along only one of the bendable faces and left to dry, a step that afforded a hybrid described as PVA/PSS//PDDA/MXene//PDDA/PSS//**2–4** (hereafter, P³//**2–4**)."

Revised text: "As shown schematically in Figure 2b and Supplementary Figure 5, nascent (as-crystallized) crystals of **2–4** were first uniformly coated with a mixture of positively charged poly(diallyldimethylammonium) (PDDA) and negatively charged poly(styrene sulfonate) (PSS) layer of ca. 650 nm thickness. The surfaces of the resulting hybrid crystals, PDDA/PSS//**2–4** (hereafter, P//**2–4**), were then coated with a mixture of (positively charged) PDDA and (negatively charged) MXene layer of ca. 200 nm thickness and described as PDDA/MXene//PDDA/PSS//**2–4** (for convenience, hereafter referred to as P²//**2–4**). Lastly, by using a needle tip, a 2 μm-thick layer of polyvinyl alcohol/poly(sodium 4-styrenesulfonate) (PVA/PSS) was deposited uniformly and rapidly along only one of the bendable faces and left to dry, a step that afforded a hybrid described as PVA/PSS//PDDA/MXene//PDDA/PSS//**2–4** (hereafter, P³//**2–4**)."

New Supplementary Figure 5. Schematic showing the method used for preparation of the hybrid crystals. (a) Preparation of P//2-4. (b) Preparation of P²//2-4. (c) Preparation of P³//2-4.

Comment: 2. *There should be more discussion on the mechanisms of different observation such as bending and waveguide.*

Response: We thank the Reviewer for the suggestion. In response to the Reviewer's comment, a discussion on the mechanisms of bending and waveguiding have been added to the text.

The following text has been added on page 5, line 132 in the revised manuscript:

Added text: "Since only one of the two wide faces of the crystal were coated with polymer, when the polymer shrinks, it gives rise to a differential strain that translates into a bending moment. This is observed as macroscopic bending of the hybrid crystal. In the hybrid crystal, the MXene functions as a photothermal converter, while the PVA/PSS layer functions as one of the two components of a bilayer strip that generates bending moment by expansion or contraction. PVA is a common hygroscopic polymer that has a low critical solubility temperature, and is well known to undergo reversible swelling via hydrogen bond formation (Supplementary Figure 6).⁴³ This bring about to response of the hybrid element to humidity, and the curvature could change with variation in humidity (Supplementary Figure 7)."

New Supplementary Figure 6. Mechanism driving the bending by photothermal effect of the hybrid crystals. The diagram shows swelling or contraction of the polymer layer induced by heating by the MXene induced by infrared light.

The following text has been added on page 10, line 240 in the revised manuscript:

Added text: “Organic crystals have been extensively studied as active optical waveguides.^{9,21,50} The principle is this property is rooted in their optical transparency and difference in refractive index with air; when one end of the crystal is excited by ultraviolet light, the excitation light is subsequently reflected and refracted continuously inside the crystal, and is eventually be transmitted to its other end.⁵¹”

Response to the comments from Reviewer #3:

General comments: *This is an interesting piece of work. The application of MXene-based materials in the field of flexible optical waveguide is a novel subject. This work provides a simple and effective way to realize the flexibility of optical waveguide on organic crystals, and this method is applicable to a variety of materials. Moreover, 2D array with high sensitivity, high and controllable degree of deformation and durability over prolonged actuation is also carried out, while the bending point of each crystal can be controlled by changing the infrared light position. Therefore I recommend this article can be considered for publication after the authors have addressed the following concerns:*

Response: We thank to the Reviewer, who apparently is a knowledgeable expert in this field, for recognizing the significance and the impact of the work presented in our manuscript, as well as for their generally positive assessment. In the revised version of the manuscript, we provide response to each of the points they have raised, as well as a description of the changes made.

Comment: 1. *The bending of organic crystals caused by thermal effect should be further analyzed theoretically. Does it involve physical changes such as phase transition or chemical changes?*

Response: We are grateful to the Reviewer for their careful inspection of the details. The bending is generally a physical phenomenon caused the generation of differential strain between two conjoined elements that have different propensity for expansion, similar to a bilayer strip. This expansion (or contraction) can be cause by either physical means, such as thermal response of the material, or chemical process such as isomerization, cyclization, etc. In either case, the result is change in length of one of the components in respect of the other component. In order to ascertain whether any chemical changes have occurred in our samples during excitation, we performed NMR analysis on the original crystals **2–4** and on the crystals **2–4** after heating them at 100 °C for 1 hour. Moreover, we performed differential scanning calorimetric (DSC) analysis on crystals **2–4**. The results confirmed that the crystals did not undergo phase transitions or chemical changes during irradiation. This data are now provided in the revised version of the manuscript.

The following text has been added on page 5, line 142 in the revised manuscript:

Added text: “To further confirm that the bending of the hybrid crystals was a result of thermally induced mechanical process and not a physical phase transition or a chemical reaction, NMR analysis of **2–4** was performed before and after heating at 100 °C for 1 hour, and the compounds were also analyzed by differential scanning calorimetry (DSC). The results ruled out phase transitions or permanent chemical changes.”

New Supplementary Figure 9. ^1H NMR spectrum of compound 2 (DMSO- d_6 , 400 MHz). (a) Spectrum of compound 2 before heating. (b) Spectrum of compound 2 after heating at 100 °C for 1 hour.

New Supplementary Figure 10. ^1H NMR spectrum of compound 3 (Chloroform- d , 400 MHz). (a) Spectrum of compound 3 before heating. (b) Spectrum of compound 3 after heating at 100 °C for 1 hour.

New Supplementary Figure 11. ¹H NMR spectrum of compound 4 (DMSO-*d*₆, 400 MHz). (a) Spectrum of compound 4 before heating. (b) Spectrum of compound 4 after heating at 100 °C for 1 hour.

New Supplementary Figure 12. Differential Scanning Calorimetric (DSC) analysis of crystals. DSC analysis of crystals **2** (a), **3** (b), and **4** (c) recorded at heating rate of 30 K min⁻¹.

Comment: 2. *Does the sample size or deposition thickness affect the performance of the 2D array?"*

Response: In order to verify this point, hybrid crystals having different sizes of the organic crystal, as well as such having same crystal size but with different thicknesses of MXene were prepared and studied. The experimental results confirmed that sample size and deposition thickness do, indeed, affect the performance of the 2D array. The new experimental results have been added to the revised version of the manuscript.

The following text has been added on page 3, line 98 in the revised manuscript:

Added text: "In addition, (PDDA/MXene)₁//PDDA/PSS//1 and (PDDA/MXene)₅//PDDA/PSS//1 were selected to demonstrate the effect of MXene thickness on the photothermal effect. As shown in Supplementary Figure 4, the temperature change of (PDDA/MXene)₅//PDDA/PSS//1 was about five-fold that of (PDDA/MXene)₁//PDDA/PSS//1 at the same power of the IR light."

New Supplementary Figure 4. Effect of MXene thickness on the photothermal effect. Photographs of the temperature change of the hybrid crystals with single layer (top) and five layers (bottom) of MXenes excited with IR light under identical conditions (408 mW).

The following text has been added on page 7, line 181 in the revised manuscript:

Added text: “Hybrid crystals with organic crystals having different size as well as with organic crystals of the same size but having different thickness of the MXene layer were also prepared to examine the effect of crystal size and MXene thickness on the performance of the 2D array. As shown in Supplementary Figure 16, four samples of P³//4 with different size of the organic crystal bent to different angles under identical experimental conditions. Similarly, two hybrid crystals having one and five layers of the MXene (PVA/PSS//PDDA/MXene)₁//PDDA/PSS//2 and PVA/PSS//PDDA/MXene)₅//PDDA/PSS//2), shown in Supplementary Figure 17, bent to different degree. These results confirmed that both the size of the organic crystal and the thickness of the MXene layer determine the performance of the 2D arrays.”

New Supplementary Figure 16. Effect of the size of the organic crystal on the bending of the hybrid crystals P³//4. Photographs are shown of different sizes of crystal P³//4 bending under irradiation with infrared light ($RH = 79\%$, 184 mW).

New Supplementary Figure 17. Effect of the thickness of the MXene layer on the bending of the hybrid crystals. Photographs of hybrid crystals PVA/PSS//((PDDA/MXene)₁//PDDA/PSS//2 and PVA/PSS//((PDDA/MXene)₅//PDDA/PSS//2 with bending changes under the same experimental conditions ($RH = 79\%$, 296 mW)

Comment:3. *The authors state that “the organic crystals have the advantages of having defect-free structures”, more characterization is needed to support this claim.”*

Response: This is an important point, and we thank the Reviewer for bringing it to our attention. Strictly speaking, no structure is completely free of defects, even that single crystals, when they are of good quality, are expected to have low number of defects, some defects included during crystallization are always present. This sentence has been removed from the text. This statement was therefore removed from the text.

The following text has been revised (page 9, line 232):

Original text: “As mentioned above, the organic crystals have the advantages of having defect-free structures, relatively low optical losses, and long-range ordered structures.”

Revised text: “As mentioned above, the organic crystals have the advantages of relatively low optical losses and long-range ordered structures.”

Comment: 4. *When it comes to fatigue and cycling, 100 times bending test is not enough. More cycles should be shown.”*

Response: We thank the Reviewer for this suggestion. To respond to the comment, additional fatigue and cycling tests were performed, and the results confirmed that the hybrid crystals maintained excellent performance after more than 1000 times. The experimental results were added to the text, and a new figure was added to the Supplementary Information.

The following text has been added (page 9, line 221) in the text:

Added text: “Moreover, fatigue and cycling tests were performed on P3//3, and the results confirmed that the performance of the hybrid crystals was retained even after 1000 cycles (Supplementary Figure 19, Supplementary Movie 9).”

The following text has been added (page 17, line 200) in the supplementary information:

Added text: “Supplementary Movie 9. Durability test of 1000 cycles of P³//3.”

New Supplementary Figure 19. Cyclability test of the bending of a hybrid crystal. Photographs of P³//3 are shown before (left) and after (right) 1000 bending cycles ($RH = 79\%$, 184 mW).

Comment: 5. *There are some typos and mistakes: the font should be uniform in Line 51, redundant words in Line 109, etc.*

Response: We thank the Reviewer for bringing these typos to our attention. The font in line 51, redundant words in line 109, and the other typos have been corrected.

The following text (page 2, line 66) has been revised:

Original text: “Herein, we report a simple and efficient method for preparation of hybrid organic crystals that are capable of responding to *infrared light*”

Revised text: “Herein, we report a simple and efficient method for preparation of hybrid organic crystals that are capable of responding to infrared light”

The following text (page 5, line 126) has been revised in the text:

Original text: “The surfaces of the resulting hybrid crystals, PDDA/PSS//2–4 (hereafter, P//2–4), were then coated with a mixture of PDDA and MXene layer of ca. 200 nm thickness, giving hybrid materials that can be described as PDDA/MXene//PDDA/PSS//2–4 (for convenience, hereafter referred to as P²//2–4).”

Revised text: “The surfaces of the resulting hybrid crystals, PDDA/PSS//2–4 (hereafter, P//2–4), were then coated with a mixture of positively charged PDDA and negatively charged MXene layer of ca. 200 nm thickness and described as PDDA/MXene//PDDA/PSS//2–4 (for convenience, hereafter referred to as P²//2–4).”

Response to the comments from Reviewer #4:

Overall Comment: *The manuscript describes the realization of rod-shaped organic crystals that exhibit a deformation response to IR light. This photoresponse is enabled by coating the crystals in a multilayer structure containing MXenes that absorb the IR irradiation. This absorption produces a local thermal response that triggers the actuation mechanism of the organic crystals. The work demonstrates this effect using a few different chemistries of organic crystals. The manuscript also shows that the location of the deformation can be controlled through the location of the illumination.*

The manuscript is convincing in its conclusion and has a clear message. I have no doubt that the crystals exhibit the deformation described by the authors. However, I have a difficult time judging the novelty and impact of the work. The manuscript is quite applied without much decision of scientific mechanisms, material design considerations, or quantitative analysis. I believe the authors should address a few specific points

Response: We are thank the Reviewer for their constructive comments. We note that the mechanism of bending of both elastic and plastic organic crystals has been thoroughly studied by various research groups over the past ten years, and is well explained in the literature. We were hesitant to include a related discussion in this manuscript and instead we cited selected references that contain more information on the mechanistic aspects. Instead, our intention was to focus on the added values of using hybrid crystals, where the thermal absorber is the MXene and the bending element is the crystal whose deformation is driven by differential expansion. This approach effectively circumvents the inherent disadvantages of organic crystals, such as, for example, the slow response of bending of photobendable crystals, which is due to the weak coupling between the changes that occur on a molecular level and the mechanical deformation that occurs on a macroscopic level. Below, we provide a point-by-point response to the Reviewer's specific comments and suggestions.

Comment: 1. *The materials should be described in much more detail. I do not believe referring to the crystals as "1" or "2" is appropriate. What are the chemical names or formula of the compounds?*

Response: We thank the Reviewer for pointing this out. The structural formulas of the compounds were added (Figure 1a, Figure 2a and Figure 5f). We apologize for referring to the crystals as "1" or "2" without giving the chemical names of the compound, which brings some lack of clarity. To respond to the comment, crystals 1–5 were named according to the order in which they appear in the text, and the actual names of the compounds were added to the text.

The following text has been revised (page 3, line 87) to include the name of the compound 1:

Original text: "In a typical case, a crystal of the organic compound 1 (Figure 1a) coated with 5 bilayers, (PDDA/MXene)₅//1, had a thickness of about 200 nm and low roughness (average roughness, 20.1 nm) (Figure 1b, c; Supplementary Figure 2).³⁹"

Revised text: "In a typical case, a crystal of the organic compound 2,2'-((1E,1'E)-1,4-phenylenebis(ethene-2,1-diyl))dibenzonitrile (for convenience, hereafter referred to as 1; Figure 1a) coated with 5 bilayers, (PDDA/MXene)₅//1, had a thickness of about 200 nm and low roughness (average roughness, 20.1 nm) (Figure 1b, c; Supplementary Figure 2).³⁹"

The following text has been also revised (page 5, line 118):

Original text: “Encouraged by these initial results with **1**, the preparation method was slightly modified and applied to centimeter-long slender elastic crystals of other three organic compounds, **2–4** in Figure 2a, which were obtained by using literature methods.^{23,41,42”}

Revised text: “Encouraged by these initial results with **1**, the preparation method was slightly modified and applied to centimeter-long slender elastic crystals of other three organic compounds, 9,10-dibromoanthracene, (*Z*)-2-([1,1'-biphenyl]-4-yl)-3-(anthracen-9-yl)acrylonitrile, and (*Z*)-3-(furan-2-yl)-2-(4-(((*E*)-2-hydroxy-5-methylbenzylidene)amino)phenyl)acrylonitrile (for convenience, hereafter named to as **2**, **3**, and **4**, respectively, Figure 2a), which were obtained by using literature methods.^{23,41,42”}

The following text has also been revised (page 11, line 260):

Original text: “A centimeter-long slender elastic crystal of compound **5** (Figure 5e, f) that has been reported earlier²⁴ was selected for the purpose.”

Revised text: “A centimeter-long slender elastic crystal of the compound (*E*)-2-(4-fluorophenyl)-3-(naphthalen-1-yl)acrylonitrile (hereafter referred to as **5**; Figure 5e, f) that has been reported earlier²⁴ was selected for the purpose.”

Comment: 2. *Do the authors have any materials characterization related to the actual materials? Have more than one of the same composition of crystal been tested? Is there any relationship between structure/quality of the crystals and the photo-mechanical response?”*

Response: The mechanical properties of crystals were characterized by three-point bending experiments, which have been now added to the text. The photothermal effect of same composition of crystal to achieve bending had been also tested, and is shown in Figure 2b, Figure 3b–j, and Supplementary Figure 16. The structure of the crystals are related to the photomechanical response while the crystal quality had a smaller effect. These details were confirmed by performing additional experiments, and are now added in the revised version of the main text and the Supplementary Information.

The following text has been added (page 3, line 90):

Added text: “In the Supplementary Figure 3, the mechanical properties of crystal **1** were tested by a three-point bending test”

New Supplementary Figure 3. Stress-strain profiles of a crystal of 1 obtained by the three-point bending test.

The following text has been added (page 5, line 146):

Added text: “In order to examine the effect of the coating with polymer and application of MXene on the mechanical properties, the stress–strain profiles of the crystals **2**, **3**, P³//**2**, and P³//**3** were compared (Supplementary Figure 13). The results confirmed that the mechanical properties of the nascent crystals were essentially retained in the hybrid crystals, with a very small change in the Young’s modulus.”

New Supplementary Figure 13. Stress-strain profiles of the native and coated crystals determined by the three-point bending test. (a) Crystal 2. (b) Hybrid crystal P³//2. (c) Crystal 3. (d) Hybrid crystal P³//3.

The following text has been modified (page 11, line 263):

Added text: “The crystal is elastic and can be bent repeatedly without breaking (Figure 5e), and has Young’s modulus of 2.49 GPa (Supplementary Figure 21)”

New Supplementary Figure 21. Stress-strain profile of a crystal of 5 obtained by three-point bending test.

The following text has been added (page 5, line 150):

Added text: “It is natural to expect that the crystal structure of the crystal determines the deformation. In line with this, as shown in Supplementary Figure 14, the bending degrees of the hybrid crystals P³//2–4 are different under identical conditions of excitation. An additional, and perhaps less obvious factor that could affect the bending, is the crystal quality. To that end, a high-quality crystal of P³//4 and a crystal of P³//3 of much poorer quality were selected and compared, as shown in Supplementary Figure 15. Both crystals bent under infrared light, which indicates that the crystal quality has a comparatively smaller effect, although quantification of this effect is not straightforward.”

New Supplementary Figure 14. Effect of crystal structure of the organic crystal on the mechanical response of the hybrid crystal. Photographs are shown of crystals P³//2 (a), P³//3 (b), and P³//4 (c) that bend under infrared light.

New Supplementary Figure 15. The effect of crystal quality on the mechanical response of the hybrid crystals. Photographs are shown of crystals P³//3 (a) and P³//4 (b) that bend under infrared light.

Comment:3. Aspects of Figure 1 are unclear. 1d shows "photographs" of the samples. What is the difference between images 0-5 in that panel? Are these the same samples as shown in the SEM images in Fig 1b? Is the AFM data in Fig 1c free from tip artifacts? The right image has streaky features that all look quite similar; do these persist when the scan direction (or sample) is rotated 90 degrees?"

Response: In the revised version, we tried to clarify these details in Figure 1. In Figure 1d, the images the figures 0 to 5 are related to the number of deposited layers of PDDA/PSS//1 and MXene. To clarify the labeling, we modified Figure 1d. The sample in Figure 1d is not the same with the one shown as an SEM image in Figure 1b. Since the samples were prepared in batches, samples from each batch were selected randomly. We concur with the reviewer on their comment on the AFM data in Figure 1c, which features pronounced trailing, resulting in similar images. To rectify this, we recorded new AFM images for (PDDA/PSS)₅//1 and (PDDA/MXene)₅//1, and the new images are added in Figure 1c. The modified Figure 1 is provided below for Reviewer's perusal.

New Figure 1. Preparation and thermal properties of the MXene-polymer-crystal hybrids. (a) Chemical structure of **1**. (b) Scanning electron microscopy (SEM) images of (PDDA/PSS)₅//**1** and the (PDDA/MXene)₅//**1** surfaces. (c) Atomic force microscopy (AFM) images of (PDDA/PSS)₅//**1** and the (PDDA/MXene)₅//**1** surfaces. (d) Photographs of different MXene layers on the (PDDA/PSS)_n//**1** surface. (n represented the number of layers of MXene) (e) UV-vis transmission spectra of (PDDA/PSS)₅//**1** and the (PDDA/MXene)₅//**1**. (f) (PDDA/MXene)₅ coated crystal temperature increase with increasing infrared optical power (ΔT , atmospheric temperature of 25 °C). (g) Time-dependent temperature increase of the (PDDA/PSS)₅//**1** and the (PDDA/MXene)₅//**1** surface under the illumination with infrared light (744 mW) at room temperature of 25 °C. The red- and light blue-shaded regions correspond to periods of light irradiation and darkness, respectively. The insets are thermographs showing temperature maps from which the temperature has been extracted.

Comment: 4. The authors state that the organic crystals have "defect-free structures". No characterization is provided to support this. Just from a basic thermodynamic standpoint, no materials are defect-free and such statements should be removed. In Fig. S2, macroscopic defects are clearly visible in the crystals"

Response: This is an important point, and we thank the Reviewer for bringing it to our attention. We agree with the Reviewer in that this is an overstatement, and any real crystal naturally includes some defects. Accordingly, this part of the sentence has been removed from the text:

Original text: “As mentioned above, the organic crystals have the advantages of having defect-free structures, relatively low optical losses, and long-range ordered structures.”

Revised text: “As mentioned above, the organic crystals have the advantages of relatively low optical losses and long-range ordered structures.”

Comment: 5. *The authors should provide details of how they synthesized the MXene. The statement in the supplemental that "The MXene was synthesized by using literature methods" is insufficient.*”

Response: We agree with the Reviewer that this is an important detail, and the synthesis of the MXene method has been added to the Supplementary Information:

Added text: “To prepare the $\text{Ti}_3\text{C}_2\text{T}_x$ MXene suspension, Ti_3AlC_2 powder was purchased from Energy Chemical. 2.0 g Ti_3AlC_2 powder was mixed with 2.0 g LiF and 30 mL 9.0 M HCl, and stirred at 35 °C for 24 hours. The mixture was washed by centrifugation at 3500 rpm several times until the pH of the supernatant was 7. Then the sediment of MXene was collected and mixed with 40 mL of deionized water. The mixture was sonicated under N_2 atmosphere for one hour and then centrifuged at 3500 rpm for one hour to obtain the MXene suspension. The MXene powder was prepared by freeze-drying of the MXene suspension (1 mg mL^{-1}).”

Reviewer comments, second round review

Reviewer #1 (Remarks to the Author):

The author has provided a thorough explanation of the issues previously raised. However, while reviewing the explanations, there are concerns about new potential problems. The authors stated that hybrid crystal would be utilized as a controllable waveguide, but all experiments regarding bending were conducted through experiments with light pulses. This approach appears to be disadvantageous for transmitting light signals in a specific direction for long period. Moreover, MXene has been reported in academia as a material with high thermal conductivity. It is expected that hybrid crystal fabricated using such a material would share this characteristic. However, there is insufficient information on how heat spreads when light is applied to a specific area for a long period of time and how the bending angle changes accordingly. The experiments in the paper examined only a specific area for a short period of time with IR light, causing bending only in the desired area. However, if IR light is applied for a long period of time, due to its unique high thermal conductivity, bending may occur not only in the desired area but in all areas. The completeness of the revised paper has improved compared to the original, and the authors have provided satisfactory answers to the previously identified issues. However, there is still uncertainty regarding whether the fabricated waveguide can perform the role of controllable wave guide they mentioned.

1. Please provide data that can demonstrate the possibility of transmitting light signals in a specific direction for an extended period of time.
2. Please provide me with data regarding hybrid crystals' behavior when exposed to IR light in a particular area over time. This data should focus on the temperature change and bending angles of the hybrid crystal, which would help reader to understand how heat is transferred through it over time and how the bending angle changes accordingly.

Reviewer #2 (Remarks to the Author):

The authors have revised the manuscript significantly by addressing most of the reviewers' comments. I recommend the acceptance of the manuscript for publication.

Reviewer #3 (Remarks to the Author):

I think the authors have addressed my concerns satisfactorily.

Reviewer #4 (Remarks to the Author):

The authors have done an admirable job of addressing the comments of all the reviewers. They have made substantial improvements to the revised manuscript, which contains significantly more data than the original submission. I recommend publication in its current form.

Response to reviewers' comments on the manuscript "Collective Motion of MXene-Polymer Hybrids with Organic Crystals Driven by Infrared Light"

Response to the comments from Reviewer #1:

General comments: *The author has provided a thorough explanation of the issues previously raised. However, while reviewing the explanations, there are concerns about new potential problems. The authors stated that hybrid crystal would be utilized as a controllable waveguide, but all experiments regarding bending were conducted through experiments with light pulses. This approach appears to be disadvantageous for transmitting light signals in a specific direction for long period. Moreover, MXene has been reported in academia as a material with high thermal conductivity. It is expected that hybrid crystal fabricated using such a material would share this characteristic. However, there is insufficient information on how heat spreads when light is applied to a specific area for a long period of time and how the bending angle changes accordingly. The experiments in the paper examined only a specific area for a short period of time with IR light, causing bending only in the desired area. However, if IR light is applied for a long period of time, due to its unique high thermal conductivity, bending may occur not only in the desired area but in all areas. The completeness of the revised paper has improved compared to the original, and the authors have provided satisfactory answers to the previously identified issues. However, there is still uncertainty regarding whether the fabricated waveguide can perform the role of controllable wave guide they mentioned.*

Response: We appreciate very much the feedback provided by the reviewer, and the additional experiments that they have requested, although these were not part of the first round of their comments. The ability of the fabricated waveguide to function as a controllable waveguide is of utmost importance for the effective transmission of optical signals. As a result, we have meticulously addressed each of the reviewer's comments in the following response.

Comment: 1. *Please provide data that can demonstrate the possibility of transmitting light signals in a specific direction for an extended period of time.*

Response: This is a very important point, and we are grateful to the Reviewer for suggesting these experiments. The hybrid crystals were placed in an environment with controlled humidity (RH = 79%). Infrared light (184 mW) was used to control the bending in a specific direction, and the optical signal transmission in a specific direction over an extended period of time was recorded. As shown in Supplementary Movie 14, the bending angle of the crystal remained constant without visible changes even after 20 minutes exposure. This experiment demonstrated the capability of the hybrid crystals to transmit light signals in specific directions over long periods of time.

To clarify this point, the following text was added on page 11, line 208 in the revised manuscript:

Added text: "As shown in Supplementary Movie 14, P³//3 transmitted the optical signal for 20 minutes without visible changes in the direction of optical transmission, confirming the capability of the hybrid crystal to transmit optical signals in a specific direction over prolonged periods of time."

Comment: 2. *Please provide me with data regarding hybrid crystals' behavior when exposed to*

IR light in a particular area over time. This data should focus on the temperature change and bending angles of the hybrid crystal, which would help reader to understand how heat is transferred through it over time and how the bending angle changes accordingly.

Response: The change in bending angle of P³//3 exposed to infrared light (184 mW) over a long time (20 mins) has been recorded (RH = 79%). In Supplementary Movie 12, the experiments confirmed that the bend angle of the hybrid crystal remained almost unchanged when exposed to IR light in a particular area over time. In the meantime, infrared camera was used to record how the hybrid crystal surface temperature changes during the excitation (20 mins). As shown in Supplementary Figures 4, 5 and Supplementary Movie 1, when the hybrid crystal was irradiated by infrared light, the heat was transferred from the irradiated area to the surrounding area (due to the very favorable thermal conductivity of MXene material), which raised the temperature of the surrounding crystal surface area. When the crystal surface temperature stabilized, the warming took a few seconds. We conclude that, overall, the heating effect of the MXene materials took place within a few seconds and did not contribute to the bending of the hybrid materials.

In order to clarify these details, the following text has been added on page 3, line 108 in the revised manuscript:

Added text: “(PDDA/MXene)₅//PDDA/PSS//1 was selected to investigate the relationship between the thermal effect caused by the MXene and the surface temperature when the hybrid crystal is irradiated by infrared light. As shown in Supplementary Figures 4 and 5 and Supplementary Movie 1, when the hybrid crystal is exposed to infrared light, heat is transferred from the irradiated area to the surroundings, due to the favorable thermal conductivity of the MXene. This raises the temperature of the surrounding crystal surface, until it becomes constant after a few seconds.”

New Supplementary Figure 4. Photothermal conversion process of the hybrid crystal.

Heat transfer across the (PDDA/MXene)₅//PDDA/PSS//1 surface over time monitored by thermal videography. The images were recorded from left to right, top to bottom, with 0.5 s interval between two images.

New Supplementary Figure 5. Photothermal conversion of the hybrid crystals. The process was monitored via the temperature change of the surface of (PDDA/MXene)₅//PDDA/PSS//1 over time.

In order to clarify this point, the following text has been added on page 10, line 265 in the revised manuscript:

Added text: “P³//3 was selected to investigate the possible variation in bending angle of the hybrid crystal over time under long-term irradiation with infrared light. As shown in Supplementary Movie 12, the bending angle of the hybrid crystal remains almost unchanged up to at least 20 minutes of exposure to infrared light.”

Response to the comments from Reviewer #2:

General comments: *The authors have revised the manuscript significantly by addressing most of the reviewers' comments. I recommend the acceptance of the manuscript for publication.*

Response: We are delighted to know that the manuscript has been revised to meet the Reviewers' expectations, and we thank the Reviewer for their constructive comments, which have led to a significant improvement in the quality of the manuscript.

Response to the comments from Reviewer #3:

General comments: *I think the authors have addressed my concerns satisfactorily.*

Response: We are grateful that the manuscript revisions have been made to the Reviewer's satisfaction, and we thank the Reviewer for their time and the constructive and encouraging comments.

Response to the comments from Reviewer #4:

Overall Comment: *The authors have done an admirable job of addressing the comments of all the reviewers. They have made substantial improvements to the revised manuscript, which contains significantly more data than the original submission. I recommend publication in its current form.*

Response: We are pleased to know that the manuscript has been revised to meet the Reviewer's expectations, and we thank the Reviewer for their constructive input which significantly improved the quality of our manuscript.